# Molecular orbital breaking in photo-mediated organosilicon Schiff base ferroelectric crystals

Zhu-Xiao Gu[1], Nan Zhang[2], Yao Zhang[2], Bin Liu[1], Huan-Huan Jiang[2], Hua-Ming Xu[2], Peng Wang[1,2], Qing Jiang [1], Ren-Gen Xiong [3] & Han-Yue Zhang [2] ✉

Ferroelectric materials, whose electrical polarization can be switched under external stimuli, have been widely used in sensors, data storage, and energy conversion. Molecular orbital breaking can result in switchable structural and physical bistability in ferroelectric materials as traditional spatial symmetry breaking does. Differently, molecular orbital breaking interprets the phase transition mechanism from the perspective of electronics and sheds new light on manipulating the physical properties of ferroelectrics. Here, we synthesize a pair of organosilicon Schiff base ferroelectric crystals, (R)- and (S)-N-(3,5-di-tert-butylbenzylidene)-1-((triphenylsilyl)oxy)ethanamine, which show optically controlled phase transition accompanying the molecular orbital breaking. The molecular orbital breaking is manifested as the breaking and reformation of covalent bonds during the phase transition process, that is, the conversion between C = N and C−O in the enol form and C−N and C = O in the keto form. This process brings about photo-mediated bistability with multiple physical channels such as dielectric, second-harmonic generation, and ferroelectric polarization. This work further explores this newly developed mechanism of ferroelectric phase transition and highlights the significance of photo-mediated ferroelectric materials for photo-controlled smart devices and bio-sensors.

Symmetry breaking, which is widely found in nature, is the key to understanding many interesting physical phenomena[1–4]. Time reversal and inversion symmetry are of great significance in the study of superconductivity[5,6], while conventional investigation of ferroelectricity is based on spatial symmetry breaking[7]. Ferroelectric materials are widely used in various applications such as data storage, sensors, and transducers because of their robust spontaneous electrical polarization[8–11]. According to Landau's phenomenological theory, they generally undergo phase transitions which can be described as spatial symmetry breaking[12,13]. Phase transition is essential for ferroelectrics

since it can bring about tunable physical properties[14]. The phase transition mechanism of ferroelectrics basically includes two types: order-disorder and displacive type. The former is usually found in some molecular ferroelectrics, while the latter is common in inorganic ceramics such as $BaTiO_3$ and $Pb(Ti, Zr)O_3$[15]. These processes are accompanied by changes in symmetric elements. For example, the structural phase transition with the Aizu notation of mm2F1 is accompanied by symmetry breaking with the loss of the two-fold screw axis and glide planes[16]. This symmetry breaking will contribute to the rearrangement of ferroic orders and changes in spontaneous

[1]Division of Sports Medicine and Adult Reconstructive Surgery, Department of Orthopedic Surgery, Nanjing Drum Tower Hospital, Affiliated Hospital of Medical School, Nanjing University, 321 Zhongshan Road, Nanjing 210008 Jiangsu, P. R. China. [2]Jiangsu Key Laboratory for Biomaterials and Devices, State Key Laboratory of Digital Medical Engineering, School of Biological Science and Medical Engineering, Southeast University, Nanjing 210009, P. R. China. [3]Ordered Matter Science Research Center, Nanchang University, Nanchang 330031, P. R. China. ✉e-mail: 101013201@seu.edu.cn

polarization, thereby achieving regulation of ferroelectric properties[7]. Symmetry breaking is definitely important for ferroelectrics; however, it may not meet all of the requirements of the increasingly diverse systems for ferroelectric compounds.

Molecular orbital breaking can help us understand some unconventional ferroelectric mechanisms from the perspective of electronics, and further provides a new approach to modulating ferroelectric properties at the electronic level especially the polarization, which is a beneficial supplement to symmetry breaking. Although there are two reports mentioned molecular orbital breaking[16,17], the investigation on this theory is still limited and lacks systematicity. We summarized and then divided molecular orbital breaking into three major categories (Fig. 1): Type 1: valence bond recombination, for which the phase transition process involves the breaking and reformation of covalent bonds and thus leads to polarization changes. For example, Liao et al.[16]. disclosed the dual breaking of molecular orbitals and spatial symmetry in a photochromic diarylethene ferroelectric material. This is the first time that this new ferroelectric mechanism–molecular orbital breaking is proposed. Type 2 is spin-state transition, for which the phase transition can be described as the variation of electron spin states and thereby changing the spontaneous polarization of materials. This can be illustrated through the [FeCo] dinuclear complex reported by Sato et al.[18]. A significant polarization change of $0.45\,\mu C\,cm^{-2}$ occurs with the conversion between high-spin and low-spin states of Fe in this complex. Type 3, energy level splitting, for which uneven electron distribution causes energy level differentiation and even molecular orbital distortion, such as the Jahn-Teller effect. For example, the thermochromic ferroelectric compound [DMeDABCO]CuCl$_4$ exhibits *mmm*F*mm*2-type ferroelectric phase transition, where the distortion of [CuCl$_4$]$^{2-}$ tetrahedron caused by energy level differentiation plays an important role during this transition[19].

Photo-mediated ferroelectric generally exhibits photo-induced structural phase transition, which is a typical material with type-1 molecular orbital breaking. Under specific wavelength illumination,

such materials absorb photons and undergo electronic transitions, which manifest as covalent bond recombination and the formation of new states[20,21]. This process is likely to be accompanied by dramatic changes in physical properties including polarization, dielectric constant, and second-harmonic generation (SHG) intensity, realizing the regulation of ferroelectric performance. Schiff bases are ideal photochromic/thermochromic materials for constructing optically controlled ferroelectrics[14,22]. Their facile synthesis, low cost, and excellent reversible photo/thermo-responsive performances make them a good platform[23]. Schiff bases present reversible tautomerism between the enol form and keto form after the UV light illumination[24]. This transition is a proton-transfer process and will bring forth two stable states with distinguished physical properties[25]. In comparison with the traditional ones driven by thermal stimuli, these materials will have promising application prospects in biomedicine since their physical properties can be mediated by light at a temperature matching the human body[26]. Additionally, Schiff bases show considerable biological activity (ability against cancer, viruses, fungus, and bacteria) and are widely used in waveguiding, drug delivery, and biocatalysis[27,28]. Therefore, Schiff-based ferroelectrics are desirable materials for biomedicine and bioelectronics.

Herein, we successfully synthesized a pair of organosilicon Schiff base ferroelectric crystals (*R*)- and (*S*)-*N*-(3,5-di-tert-butylbenzylidene)-1-((triphenylsilyl)oxy)ethanamine (*R*- and *S*-SEA-Si). These organosilicon compounds crystallize in the *P*2$_1$ space group and both of them exhibit reversible photoisomerization accompanied by molecular orbital breaking. The switching of enol and keto forms triggered by the UV/vis light can be illustrated by the change in the hybridization of N and O atoms and the subsequent breaking and reformation of their corresponding covalent bonds (C = N and C–O bonds and C–N and C = O bonds). In addition to the typical ferroelectric bistability modulated by the specific electric/light field, *R*- and *S*-SEA-Si show photomediated physical properties in multiple channels including dielectric and SHG properties. As expected, these organosilicon Schiff base

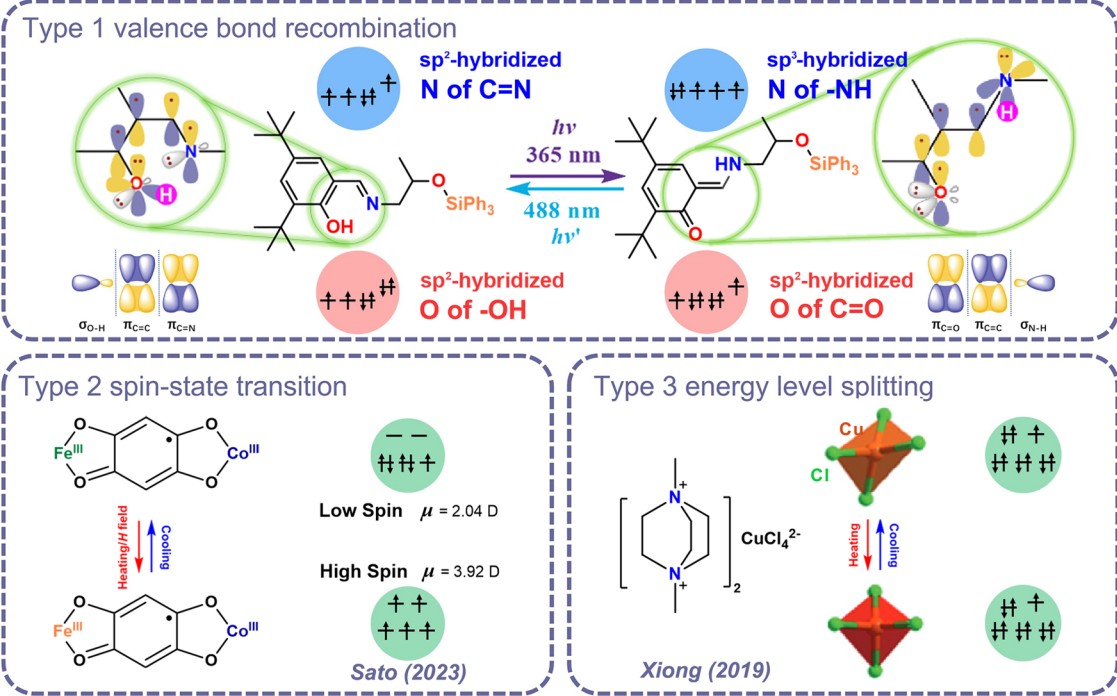

**Fig. 1 | Schematic of three types of molecular orbital breaking.** Type 1: valence bond recombination; type 2: spin-state transition; type 3: energy level splitting.

compounds show excellent biocompatibility and promotion of cell proliferation in MC3T3-E1 cells confirmed by cell viability and adhesion tests. These intriguing features enable the multi-channel modulation in physical properties without the changing of temperature, thus making *R*- and *S*-SEA-Si beneficial supplements to traditional ferroelectrics whose switchable properties are generally realized by thermodynamics. This finding extends the ferroelectric mechanism for the optically-controlled ferroelectrics and inspires further exploration of their applications in biomedicine.

## Results

### Characterization of structures of *R*- and *S*-SEA-Si

Introducing homochirality is an effective strategy to induce ferroelectricity since homochiral compounds are more likely to crystallize in five chiral-polar space groups[29,30]. Furthermore, the introduction of organosilicon groups may combine the excellent mechanical properties of organosilicon materials and the electromechanical performance of ferroelectrics, making these new materials more adaptable for bioelectronic devices[31,32]. Based on the mentioned chemical design strategies, we obtained a series of organosilicon Schiff base compounds *R*-, *S*-, and *Rac*-SEA-Si through solution processing. Their phase purities and thermal stabilities (up to 579 K) were proved by powder X-ray diffraction (PXRD) (Supplementary Fig. 1) and thermogravimetric analysis (TGA) (Supplementary Fig. 2). Differential scanning calorimetry (DSC) analyses indicate the absence of thermodynamic structural phase transition before their melting points (378 K for the enantiomers and 365 K for the racemic compound) (Supplementary Fig. 3). The enantiomeric feature was investigated by the circular dichroism (CD) spectra. The CD spectra of the enantiomers exhibit a mirror-like relationship (Supplementary Fig. 4). The SHG measurement was carried out to explore crystal symmetry (Supplementary Fig. 5). The single-crystal structures of the synthesized compounds were determined by single-crystal X-ray diffraction (XRD) measurements at 223 K. The racemic compound adopted the centrosymmetric $P2_1/c$ space group, and the asymmetry unit included four molecules. As expected, the enantiomers both crystallized in the monoclinic chiral-polar $P2_1$ space group (Supplementary Table 1), which conforms to the essential condition of ferroelectricity. The asymmetry unit contained two corresponding molecules (Fig. 2a). The distance between the O atom of the hydroxyl group and the N atom is 2.570 and 2.569 Å for *R*- and *S*-SEA-Si, respectively. Resembling a typical Schiff base compound saliclideneaniline, the intramolecular O−H···N hydrogen bonding forms the pseudo-six-member ring that indicates the *cis*-enol form. As shown in Fig. 2b, the homochiral SEA-Si molecules aligned symmetrically along the $2_1$ screw axis in a head-to-tail manner to form the packing crystal. C−O and C−N bonds of these molecules point to the same side with an ordered arrangement, resulting in the generation of spontaneous polarization along the *b*-axis. The adjacent molecules are staggered and stacked in the ab plane. The distance between adjacent N atoms of *S*-SEA-Si is 10.470 Å (Fig. 2c). There is enough space for the free rotation movement and photoisomerization from enol to *trans*-keto form because of the introduction of the large triphenylsilyl group as well as the staggered arrangement. As expected, as shown in the inset of Fig. 3a, these three compounds show an apparent color change from light yellow to orange under the irradiation of UV light (365 nm), and this process is reversible under the irradiation of visible light (488 nm). However, because of the low yield of the keto configuration since photoisomerization only occurs on the surface and not in the core of the crystal, we cannot obtain the single-crystal structures of these compounds in the keto forms.

To investigate the enol-keto phototransformation, we carried out UV-Vis absorption measurements before and after UV irradiation (Fig. 3a). The enantiomers showed similar UV absorption behavior, and the clear absorption edges at around 510 nm were noticed, corresponding to the yellow color of the enol form. After the irradiation of

365 nm light, the UV−vis spectra showed an obvious change with the absorption edge extending to around 550 nm, which is in accordance with the orange color of the *trans*-keto molecular configuration. Thus, this photochromic phenomenon could be attributed to the red shift of the optical absorption edge, resulting from the corresponding variation of electronic transition in *R*- and *S*-SEA-Si. We have investigated the stability of the *trans*-keto form of *R*-SEA-Si through time-dependent solid-state UV-vis spectra and TGA measurements (Supplementary Figs. 6−7). This photochromic behavior was also investigated through infrared (IR) spectroscopy (Fig. 3b). As expected, an obvious absorption peak at around 3415 cm$^{-1}$ emerged after UV irradiation. Further frequency calculation indicates that only the *trans*-keto form shows a strong absorption at around 3577 cm$^{-1}$ (Supplementary Fig. 8)[33]. This can be assigned to the free N−H stretching vibration, which is close to the experimental results. Accordingly, the photochromic behavior observed in *R*- and *S*-SEA-Si compounds is the transformation from enol to *trans*-keto forms.

From the perspective of molecular orbital breaking, this photo-induced enol to *trans*-keto transformation process involves the breaking of old bonds (σ bond of O−H and π bond of C = N) and the generation of new bonds (σ bond of N−H and π bond of C = O) (Fig. 1). During this process, the sp$^2$-hybridized N in the C = N group transforms into the sp$^3$-hybridized N in the C−NH. Specifically, N atoms in the imines are inequivalent sp$^2$ hybridized, where the p-orbitals are involved to form the π-bonds, two of the sp$^2$-orbitals form the σ bonds with the adjacent C atoms, and the remaining sp$^2$-orbital has a lone pair of electrons. After this phase transition, N atoms in the amino groups adopt the equivalent sp$^3$ hybridization. Three of these orbitals form the σ bonds with the 2p-orbitals of two C atoms and the 1 s orbitals of one H atom, respectively. There is a lone pair of electrons in the remaining orbital. With regard to the O atoms, although the sp$^2$-hybridized O atoms remain unchanged during the phase transition process, their molecular orbits change greatly. Two sp$^2$-orbits form two σ bonds with H atoms forming the enol form. After the phase transition, one sp$^2$ orbit and one p orbit respectively form one σ and π bond in the C = O group resulting in the keto configuration (Fig. 1).

To further understand the molecular orbital breaking, the highest occupied molecular orbital (HOMO) and lowest unoccupied molecular orbital (LUMO) of *R*- and *S*-SEA-Si molecules in enol form were calculated (Supplementary Fig. 9 and Fig. 3c and d). From the HOMO of the *R*- and *S*-SEA-Si molecules, the electron density is mainly distributed on the benzene ring from salicylaldehyde moiety and neighboring N atom as well as the O atom. The LUMO has a shift to the C − N direction in comparison with the HOMO. The calculated energy gap between HOMO and LUMO is 0.1628 Hartree, corresponding to an energy of 4.43 eV. When the molecule absorbs a UV photon, this typically causes an electronic transition in which an electron is promoted to an antibonding orbital[34,35]. Considering the LUMO distribution, the proton prefers to transfer from the O atom to the N atom. Thus, according to the principle of orbital symmetry matching as well as the variation of radial distribution of orbitals of O and N atoms, molecular orbital breaking occurs accompanied by the electron transfer in these organosilicon compounds.

### Physical properties of *R*- and *S*-SEA-Si

Compared to conventional ferroelectrics whose physical properties are mediated by thermodynamics, photo-mediated ferroelectric materials can realize the modulation in a moderate environment (Fig. 4a). The temperature-controlled ferroelectrics usually have extremely high phase transition temperature that exceeds the tolerance of human beings. The high temperature makes the modulation of physical properties a threat to human safety. The modulation of photo-mediated ferroelectrics can be realized by irradiation of specific light in the range of temperature that people can adapt to, making them promising candidates for applications in personal health management,

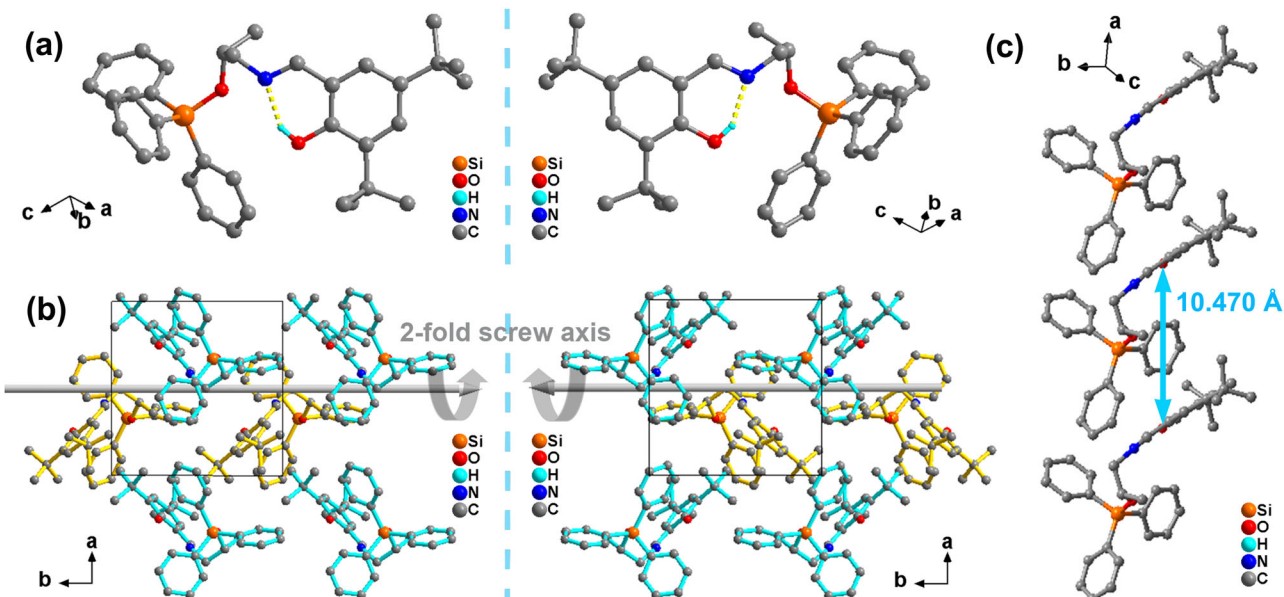

**Fig. 2 | Crystal structure of *R*- and *S*-SEA-Si. a** Molecular structures of *R*- and *S*-SEA-Si. **b** Packing pattern of *R*- and *S*-SEA-Si, where blue color molecules and yellow color molecules are symmetrical about $2_1$ screw axes (represented by the gray arrow) along the *c*-axis at 223 K. **c** Packing view of *S*-SEA-Si. The dashed lines denote the distance between the adjacent N atoms. H atoms are partly omitted for clarity.

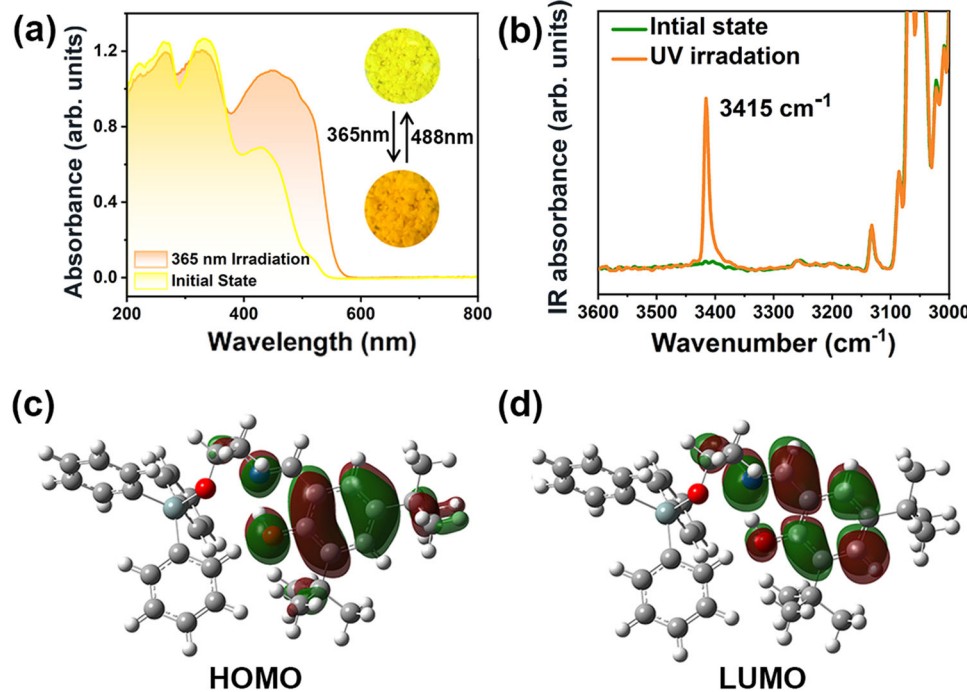

**Fig. 3 | The spectra and orbital structures of *R*- and *S*-SEA-Si.** Experimental UV-Vis (**a**) and IR absorption (**b**) spectra of *R*-SEA-Si under ambient conditions and after UV irradiation. HOMO (**c**) and LUMO (**d**) of the *R*-SEA-Si molecule.

sports detection, electronic skin, and other fields[36]. Apart from the ferroelectric bistability, these organosilicon compounds have photomediated bistability in multiple channels including dielectric, ferroelectric polarization, and SHG properties. With exposure to UV light, the real part $\varepsilon'$ of the dielectric constant of compound *S*-SEA-Si changed from about 3.1 in the enol form to 4.3 in the keto form at 750 kHz (Fig. 4b). Then, the $\varepsilon'$ can swiftly return to its original state after irradiation of 488 nm light. Accordingly, this dielectric switching between the high-dielectric state in the keto form and the low-dielectric state in

the enol form resulted from the light-triggered enol-keto transformation.

SHG effect is closely related to the polarization states[37]. Figure 4c exhibited the switching of SHG intensity of *S*-SEA-Si under UV/visible irradiation. The SHG intensity changes following the state change caused by the photoisomerization between the enol/keto form under UV/visible, respectively. Specifically, the high/low-SHG state corresponds to the enol/keto form. Such bistable SHG response is desirable for sensors and data storage in some multifunctional optical devices[38].

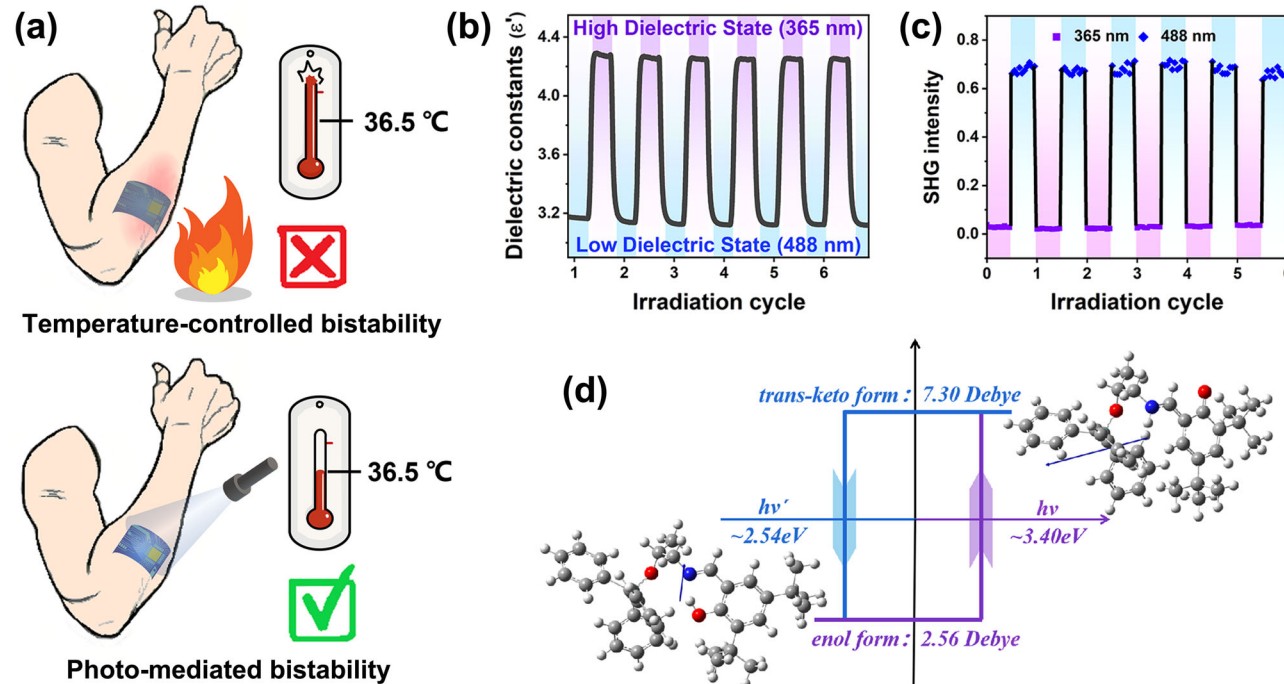

**Fig. 4 | Photo-mediated bistability of *R*-SEA-Si in three physical channels: dielectricity, SHG intensity, and ferroelectricity. a** Comparison of the conventional temperature-controlled and photo-mediated bistability. **b** The real part $\varepsilon'$ at room temperature between the irradiation of 365 and 488 nm. **c** SHG intensity between the irradiation of 365 and 488 nm at room temperature. **d** Schematic diagram of the molecular conformation transformation. The blue arrow indicates the direction of the molecular dipole.

To estimate the ferroelectric polarization of these compounds, we first calculated the vector sum of the dipole moments of the molecules in the unit cell. The dipole moment of *R*- and *S*-SEA-Si molecules is 2.58 Debye. Considering the arrangement and dipole orientation of the molecular cell, the total polarization value is estimated to be about 1.17 μC cm⁻² (along the *b*-axis). The polarization along the *a* and *c* axes equals zero, satisfying the symmetry requirement of space group *P*2₁. To compare the light-induced change of dipole moments in *R/S*-SEA-Si molecules, we thus calculated the dipole moment of enol and *trans*-keto configurations. The molecular dipole moment changed from 2.58 Debye to 7.30 Debye, corresponding to the transition from enol to *trans*-keto forms. This optically driven ferroelectric bistability can be described with a schematic diagram of the hysteresis loop, where the coercive field is the energy required for the photoisomerization (Fig. 4d).

Piezoelectric force microscopy (PFM) is a powerful tool for studying the statics and dynamics of ferroelectric polarization, which can nondestructively visualize and manipulate a microscale domain structure with high spatial resolution[39,40]. We thus used PFM to investigate the ferroelectricity of the crystalline thin film of *R*-SEA-Si. The domain does not overlap with its topography (Fig. 5a–c). Further, the box-in-box switched domain patterns can be written on its thin film with the application of opposite voltages of ±150 V (Fig. 5d–g). This offers solid evidence for electrically ferroelectric polarization switching. PFM switching spectroscopy was also performed to measure the corresponding local hysteresis loops. Figure 5h, i shows the PFM amplitude and phase loops for *R*-SEA-Si at the initial state and after UV light illumination of 365 nm. The butterfly-shaped amplitude loops and hysteresis phase loops confirmed the electrically controlled ferroelectric switching in both enol and keto forms.

It is known that some Schiff base ferroelectric compounds show optically controlled ferroelectric polarization switching. To investigate the optical control of the domains in *R*-SEA-Si, the experiment was conducted as follows. Firstly, we imaged the out-of-plane PFM phase and amplitude image of the initial domain pattern in the dark, corresponding to the enol state (Figs. 6a and d). Then, the UV light irradiation of 365 nm was exposed to the surface of the *R*-SEA-Si crystalline thin film, and we in situ recorded the PFM images. As shown in Figs. 6b and e, the optical poling domain structure of *trans*-keto state was obtained upon UV light irradiation of 365 nm, resulting in a uniform brown single-domain state. After that, the visible light of 488 nm was exposed to the sample surface (Figs. 6c and f). The brown domain can be recovered to light green domains after laser exposure of 488 nm, corresponding to the enol state. The light sources utilized here are LED lamps with a rated power of 3000 mW and 60 mW for wavelengths of 365 nm and 488 nm, respectively.

## Biocompatibility of *R*-SEA-Si

Generally, organosilicon compounds have satisfying biocompatibility and physiological inertia. To investigate the biocompatibility of these compounds, we carried out the in vitro biocompatibility assessment of as-prepared *R*-SEA-Si using MC3T3-E1 cells as model, the process of which was schematically shown in Fig. 7a. As expected, *R*-SEA-Si exhibited no cytotoxicity towards MC3T3-E1 cells after 24 and 48 h of co-incubation (Fig. 7b). Surprisingly, significant promotion effect of *R*-SEA-Si on cell proliferation of MC3T3-E1 cells could be observed with the cell viability of 129.9 ± 8.2 % and 153.1 ± 6.3 % compared with the control group respectively. To further visually evaluate the potential cytotoxicity of *R*-SEA-Si towards MC3T3-E1 cells, live/dead staining, a frequently used method to distinguish between viable (live) and non-viable (dead) cells was employed. The staining procedure involves two fluorescent dyes that are selectively taken up by either live or dead cells[41,42]. As illustrated in Fig. 7c, nearly all the cells were alive (green fluorescence) after treated with leach liquor of *R*-SEA-Si for 24 and 48 h. Quantification of live/dead staining was assessed using ImageJ and the result was demonstrated in Fig. 7d, in which the proportions of viable cells were 99.41% for the control group, 99.77% for 24 h of the *R*-SEA-Si-treated group and

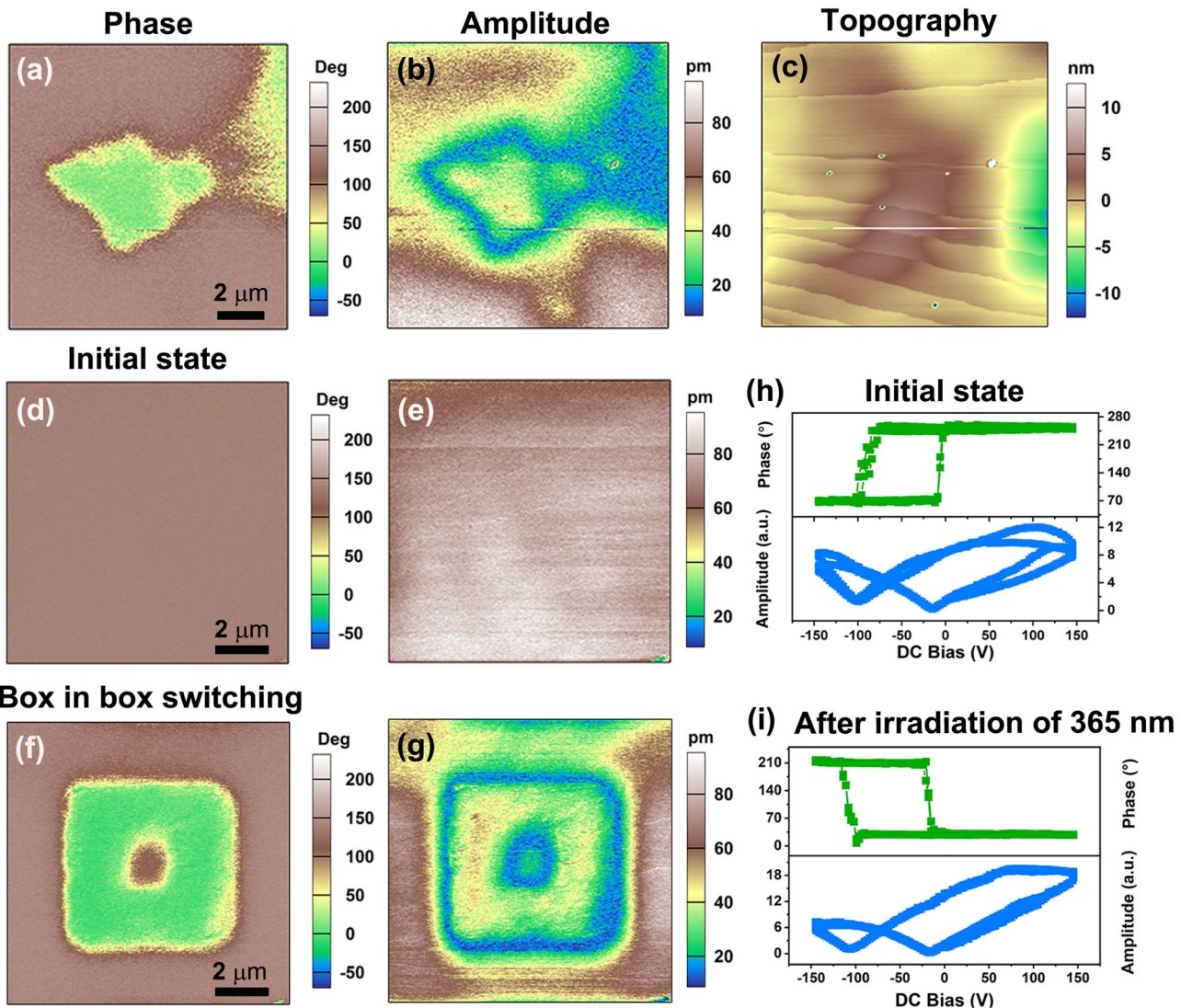

**Fig. 5 | Ferroelectricity of *R*-SEA-Si.** PFM phase (**a**), amplitude (**b**), and topography (**c**) images. Electric field-induced polarization switching on the crystalline thin film of *R*-SEA-Si. PFM phase (**d**) and (**e**) amplitude images in the initial state. **f** PFM phase and (**g**) amplitude images after applying a tip voltage of +150 V on the central region and then −150 V on a smaller central region. Phase and amplitude loops obtained by PFM switching spectroscopy measurements under dark conditions (**h**) and irradiation of 365 nm (**i**).

99.89% for 48 h of *R*-SEA-Si-treated group, which is consistent with the cell viability results. In addition, the effect of *R*-SEA-Si on cell adhesion of MC3T3-E1 cells was detected using phalloidine/DAPI staining, which is a common method to visualize the actin cytoskeleton and the nucleus of cells under the observation of fluorescence microscopy[43]. As shown in Fig. 7e, MC3T3-E1 cells spread out with their filopodia extended after treated with leach liquor of *R*-SEA-Si, indicating that *R*-SEA-Si did not affect cell adhesion of MC3T3-E1 cells. To further investigate the potential biological toxicity of *R*-SEA-Si, human bone marrow-derived mesenchymal stem cells (hBMSCs) were utilized. Following a 48 h exposure to the leach liquor of *R*-SEA-Si, hBMSCs exhibited robust cell viability (Supplementary Fig. 10), with no discernible effects on cell proliferation (Supplementary Fig. 11) or cell adhesion (Supplementary Fig. 12). In addition, the neurotoxin of *R*-SEA-Si towards rat neural stem cells (rNSCs) was further investigated. Immunostaining images for anti-beta III tubulin (Tuj-1) and anti-glial fibrillary acidic protein (GFAP) demonstrated that rNSCs treated with the leach liquor of *R*-SEA-Si could differentiate normally as in the control group (Supplementary Fig. 13). The in vivo biosafety of *R*-SEA-Si was further investigated by subcutaneously implantation of *R*-SEA-Si (10 mg) into ICR mice for 3 days, the process of which was schematically shown in Fig. 7f. The analysis of serum parameters, including aspartate aminotransferase (AST), urea, and cholesterol (CHO), all fell within the normal range (Fig. 7g). Furthermore, neither the skin tissue directly overlying the *R*-SEA-Si implant site (Fig. 7h) nor the major organs (Fig. 7i) displayed observable inflammatory cell infiltration or pathological characteristics. Taken together, we undoubtedly confirmed that *R*-SEA-Si was biocompatible towards MC3T3-E1 cells and hBMSCs without affecting their cell viability and adhesion. These features highlight the great potential of organosilicon ferroelectrics for biomedical and human-compatible applications.

## Discussion

We present a pair of organosilicon Schiff base ferroelectric crystals, which show photo-mediated bistable characteristics in dielectric, SHG, and polarization. The bistable switching is realized through the reversible enol-keto photoisomerization. The mechanism of this process and the accompanied molecular orbital breaking are thoroughly studied. The variation of the orbits of N and O atoms and the

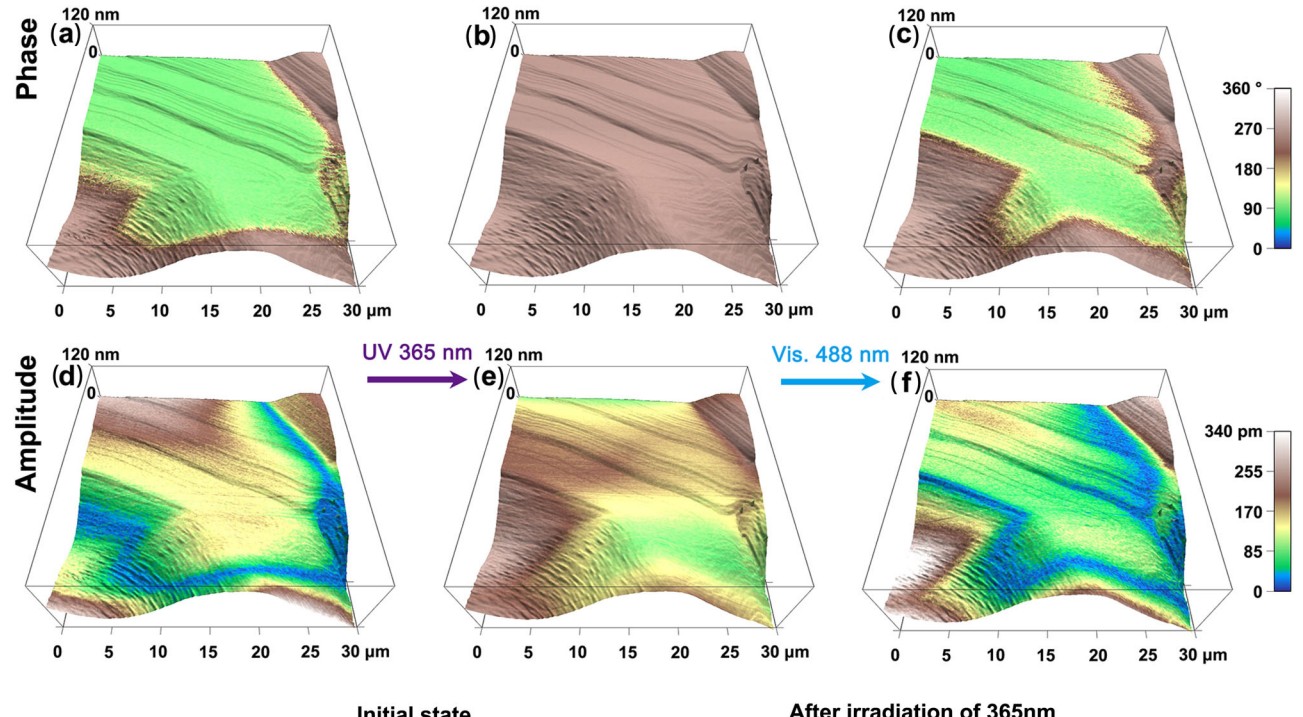

**Fig. 6 | Light-mediated manipulation of domains on the thin film of *R*-SEA-Si.** PFM phase and amplitude images for the acquisition of the initial state (**a** and **d**), after UV light irradiation of 365 nm (**b** and **e**), and after visible light irradiation of 488 nm (**c** and **f**).

breaking and recombination of covalent bonds support the photo-mediated transition. Contrary to the traditional ferroelectrics, the control of physical properties for *R*- and *S*-SEA-Si can be achieved by simple photoirradiation instead of temperature change. The ferroelectric behaviors as well as the optically controlled ferroelectric polarization switching of these compounds are confirmed by PFM. Besides, these organosilicon compounds not only show good in vitro biocompatibility as well as a significant promotion effect on cell proliferation but also exhibit excellent in vivo biosafety when subcutaneously implanted into mice. Combined with the unique photomediated ferroelectricity and satisfying biocompatibility, *R*- and *S*-SEA-Si are desirable candidates for the application of bioelectronics such as sensors, actuators, soft robotics, and optical data memory[44].

## Methods

### Synthesis and preparation of *R*- and *S*-SEA-Si

A solution of chlorotriphenylsilane (20 mmol, 5.90 g) in dry dichloromethane (12 mL) was added by drops to dry dichloromethane (12 mL) solution of corresponding isopropanolamine (20 mmol, 1.50 g), and triethylamine (40 mmol, 5.57 mL) under nitrogen atmosphere and the mixture was stirred for 12 h at room temperature. Ammonium chloride saturated aqueous solution (12 mL) was added to the reaction mixture and extracted with dichloromethane (3 × 36 mL), followed by washing with saturated sodium chloride aqueous solution. The organic phase was mixed and dried with anhydrous magnesium sulfate, filtered, and the solvent evaporated under reduced pressure to obtain the final product.

The product (10 mmol, 3.34 g) obtained in the previous step and 3,5-di-tert-butyl-2-hydroxybenzaldehyde (10 mmol, 2.34 g) were dissolved in toluene (40 mL) in a 100 mL round-bottomed flask. The reaction was stirred and refluxed at 120 °C overnight, and then purified by silica gel column chromatography (petroleum ether: ethyl acetate = 3:1). Yellow crystals were obtained by recrystallization in the mixing methanol and acetonitrile solvent.

### Single-crystal XRD and PXRD measurements

Crystallographic data were collected using a Rigaku Saturn 924 diffractometer equipped with a temperature control device, by using Cu Kα ($\lambda = 1.54184$ Å) radiation. Data processing including empirical absorption correction, cell refinement, and data reduction was performed using the Crystal Clear software package.

The data collection and structure refinement of these crystals are summarized in Supplementary Table 1. PXRD data were measured using a Rigaku D/MAX 2000 PC X-ray diffraction system with Cu Kα radiation in the $2\theta$ range of 3°–40° with a step size of 0.02°.

### DSC, TGA, and SHG measurements

DSC measurements were performed on a PerkinElmer Diamond DSC under a nitrogen atmosphere in aluminum crucibles with a heating or cooling rate of 10 K min⁻¹. The TGA measurement of *S*-SEA-Si, *R*-SEA-Si, and *Rac*-SEA-Si were performed on a PerkinElmer TGA 8000 at a heating rate of 30 K min⁻¹ in the atmosphere.

An unexpanded laser beam with low divergence (pulsed Nd:YAG at a wavelength of 1064 nm, 5 ns pulse duration, 1.6 MW peak power, 10 Hz repetition rate) was used for SHG experiments. The instrument model is Ins 1210058, INSTEC Instruments, while the laser is Vibrant 355 II, OPOTEK.

### UV-vis absorption spectra

UV–vis absorbance spectroscopy was measured on thin film and powder samples by using Shimadzu (Tokyo, Japan) UV-3600 Plus spectrophotometer equipped with ISR-603 integrating sphere at room temperature, respectively. BaSO₄ was used as a 100% reflectance reference. The thin film was detected in transmission mode, and the powder sample was detected in diffuse reflection mode.

### CD spectra and IR measurements

The potassium bromide (KBr) tablets method was used for CD measurements in transmission mode. The CD spectra are recorded with

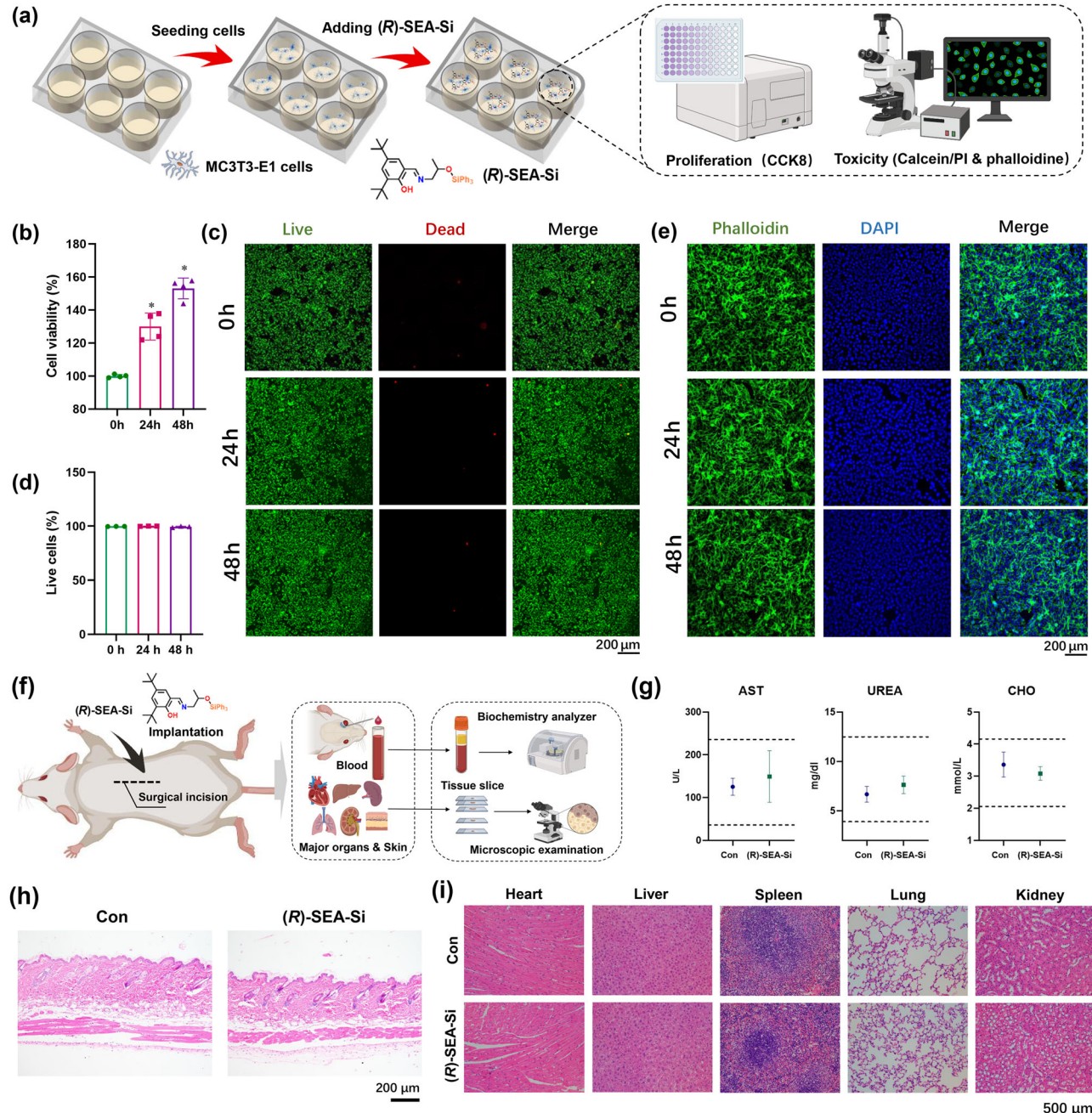

**Fig. 7 | Biocompatibility assessment of *R*-SEA-Si. a** Schematic of in vitro biocompatibility assessment of *R*-SEA-Si towards MC3T3-E1 cells. **b** Cell viability of MC3T3-E1 cells after incubation with leach liquor of *R*-SEA-Si for 24 and 48 h respectively. The data was presented as mean values +/− SD. *P* < 0.05 was considered statistically significant, which was marked with the * symbol. **c** Representative scan of live/dead staining of MC3T3-E1 cells after treated with leach liquor of *R*-SEA-Si for 24 and 48 h, green and red labeled cells represent living and dead cells, respectively. **d** Quantification of live/dead staining in Fig. 7c using ImageJ. The data was presented as mean values +/− SD. **e** Representative scan of phalloidine/DAPI staining of MC3T3-E1 cells after treated with leach liquor of *R*-SEA-Si for 24 and 48 h, green and blue labeled cells represent cytoskeleton and nucleus

of cells, respectively. 3 independent biological replicates were conducted. **f** Schematic of in vivo biosafety assessment of *R*-SEA-Si by subcutaneously implantation of *R*-SEA-Si (10 mg) into ICR mice. **g** Serum biochemical test results from control and *R*-SEA-Si administered mice at 3 days after subcutaneously implantation (*n* = 5 per group). The data was presented as mean values +/- SD. **h** H&E staining images of mouse skin tissue directly above the implanted *R*-SEA-Si. 3 independent biological replicates were conducted. **i** H&E staining images of major organs collected from control and *R*-SEA-Si administered mice at 3 days after subcutaneously implantation. 3 independent biological replicates were conducted. Figure 7(a) and (f), created with BioRender.com, released under a Creative Commons Attribution-NonCommercial-NoDerivs 4.0 International license.

---

JASCO J-1700. The KBr tablets method was used for IR measurements. The IR spectra are recorded with the PMA-50 module.

## Calculation condition
The HOMO, LUMO, VCD, and IR are calculated at b3lyp/6-31 G(d) level with Gaussian 09 software. We constructed enol, *cis*-keto, and *trans*-keto configurations based on the experimentally measured single-crystal XRD structure.

## Dielectric measurements
The dielectric measurements were carried out on a Tonghui TH2828A impedance analyzer. The capacitor for dielectric

measurement was prepared by melting the sample between two ITO transparent glasses and being recrystallized at room temperature to form a sandwich-like structure. The area of electrodes and the thickness of the film are ~90 mm$^2$ and 30 μm, respectively. In the photo-triggered dielectric measurement, the capacitor was alternately illuminated by the 365 nm and 488 nm light from two sides in the dark.

## PFM characterization

The PFM measurement was carried out on a commercial piezo-response force microscope (Oxford instrument, Cypher ES) with a high-voltage package, in-situ heating stage, and custom-designed light sources. Conductive Pt/Ir-coated silicon probes (EFM, Nanoworld) were used for domain imaging and polarization switching studies, with a nominal spring constant of ~2.8 nN/nm and a free-air resonance frequency of ~75 kHz. The thickness of the thin film of *R*-SEA-Si is about 1.53 μm. We performed the PFM experiments at contact resonance. The typical drive frequency was in the range of 300–420 kHz for vertical PFM images and 600–720 kHz for lateral PFM images, depending on the contact resonant frequency.

## In vitro biocompatibility assessment

Mouse calvarial pre-osteoblast cell line MC3T3-E1 cells and human bone marrow-derived mesenchymal stem cells (hBMSCs) were obtained from the Institute of Life Science Cell Culture Center (Shanghai, China). Rat neural stem cells were isolated according to our previous study[45].

Cells were cultured with leach liquor of *R*-SEA-Si. After 24- and 48 h of incubation, the cells were washed with phosphate buffer saline (PBS) for three times. After that, 10 μL of cell counting Kit-8 (CCK-8; Bimake, China) was added. After 1 h of incubation, the cell viability was measured using a microplate reader according to optical density at 450 nm. For live/dead staining, the cells were cultured with leach liquor of *R*-SEA-Si for 24- and 48 h and subsequently washed with PBS for 3 times. After that, the cells were stained with calcein-AM/PI (Solarbio, China) according to the manufacturer's instructions. The stained cells were observed using a laser scanning confocal microscope (Olympus 141 FV3000, Tokyo, Japan). For phalloidine/DAPI (4',6-diamidino-2-phenylindole) staining (Sigma-Aldrich, USA), the cells were cultured with leach liquor of *R*-SEA-Si for 24 and 48 h and washed with PBS for 3 times. Subsequently, the cells were fixed with 4% paraformaldehyde, treated with 0.1% Triton X-100 and stained with phalloidin and DAPI according to the manufacturer's instructions. The stained cells were observed using a laser scanning confocal microscope (Olympus 141 FV3000, Tokyo, Japan).

## In vivo biosafety assessment

24 male ICR mice (8 weeks) were obtained from the Laboratory Animal Center of Nanjing Drum Tower Hospital, Affiliated Hospital of Medical School, Nanjing University (China). The ICR mice were housed in a standard environment with a temperature of 24–26 °C, humidity of 70–75%, a 12 h light–dark cycle, typical laboratory meals. All the experimental protocols in this study were approved by the Committee of Nanjing Drum Tower Hospital, Affiliated Hospital of Medical School, Nanjing University. A 10 milligram dose of *R*-SEA-Si was subcutaneously implanted into the beneath of the mouse dorsal skin for 3 days. After that, mouse serum was collected for serum biochemical analysis. Additionally, major organs (heart, liver, spleen, lung, and kidney) as well as the skin directly above the *R*-SEA-Si implant site were harvested for histological examination.

## Reporting summary

Further information on research design is available in the Nature Portfolio Reporting Summary linked to this article.

## Data availability

All data are in a publicly accessible repository. The crystal structures generated in this study have been deposited in the Cambridge Crystallographic Data Centre under accession code CCDC: 2301143-2301145 and can be obtained free of charge from the CCDC via www. ccdc.cam.ac.uk/data_request/cif.

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

## Acknowledgements

This work was supported by the Seventh Youth Elite Scientist Sponsorship Program by the China Association for Science and Technology (H.-Y.Z.), the National Natural Science Foundation of China Grant No 12304005 (H.-Y.Z.), 32271413 (P.W.), the Natural Science Foundation of Jiangsu Province Grant No BK20230809 (H.-Y.Z.), the Zhishan Young Scholars Program of Southeast University Grant No 2242024RCB0014 (H.-Y.Z.) and the National Basic Research Program of China Grant No 2021YFA1201404 (P.W.).

## Author contributions

Z.-X.G., N.Z., and Y.Z. performed the experimental studies. Y.Z. and H.-Y.Z. carried out the analysis. Z.-X.G., B.L., P.W., and Q.J. performed the in vitro biocompatibility and in vivo biosafety assay. Z.-X.G., N.Z., H.-H.J., H.-M.X., P.W. and H.-Y.Z. wrote the manuscript. R.-G.X. offered auxiliary supervision the work. All authors commented on the manuscript.

## Competing interests

The authors declare no competing interests.
