## [Peer Review File · Nature Communications]

Molecular Orbital Breaking in Photo-Mediated Organosilicon Schiff base Ferroelectric CrystalsEditorial Note: Parts of this Peer Review File have been redacted as indicated to remove third-party material where no permission to publish could be obtained.

REVIEWER COMMENTS

Reviewer #1 (Remarks to the Author):

In this study, the authors report the synthesis of a pair of organosilicon Schiff base ferroelectric crystals, (R)- and (S)-N-(3,5-di-tert-butylbenzylidene)-1-((triphenylsilyl)oxy)ethanamine, which exhibit optically controlled phase transitions involving molecular orbital breaking. The cleavage of molecular orbitals indicates the cleavage and reconfiguration of covalent bonds (C=N and C=O conversion in the enol form and C=N and C=O conversion in the keto form) during the phase transition. In particular, the authors show that these organosilicon Schiff base compounds exhibit excellent biocompatibility and promote cell proliferation of MC3T3-E1 cells, as confirmed by cell viability and adhesion studies. I found this process reported by the authors interesting because it could lead to light-mediated bistability with multiple physical channels, including dielectricity, second harmonic generation, and ferroelectric polarization. On the other hand, I would like to request a clearer description of the following points.

1. I thought the biocompatibility of organosilicon Schiff base compounds was an important result in this paper. The authors show their results in Figure 6, which I thought was an important result. On the other hand, biocompatibility testing is defined by international standards and is performed in accordance with ISO10993-5 and ISO10993-6. I would like to ask the authors to describe whether their biocompatibility tests are consistent with these results.

2. The authors' material is shown to have two states, with covalent bond cleavage and reconstitution during the phase transition process (Figure 2). On the other hand, the stability in each state has not been sufficiently demonstrated experimentally. I believe it is necessary to demonstrate stability in each state, such as stability over time, stability under ambient light in the atmosphere and other environments, and stability with respect to

temperature.

3. As a non-academic comment, it is unnatural for the Discussion chapter to begin with "In conclusion," which is not appropriate. In this paper, the Discussion is not sufficiently thorough. In particular, I think that the novelty should be clearly indicated again in terms of what points are novel compared to previous organosilicon Schiff base ferroelectric crystals. One perspective would be a novel application utilizing biocompatibility, but I do not believe that the biocompatibility tests have been shown from various experimental results. I think that standard experimental results to demonstrate biocompatibility should be presented, and I would like to request the presentation of experimental results that many researchers such as ISO10993, HE, Iba1, GFAP, etc. have shown as biocompatibility tests.

Reviewer #2 (Remarks to the Author):

The article describes the investigation of molecular orbital breaking in organosilicon Schiff based molecular crystals as a novel foundation for ferroelectricity which can be induced by light stimulation. The orbital breaking mechanism involves the shift of a double bond in the compound from the C-N to the C-O bond. The primary interest seems to be the materials biocompatibility and thus the use of this light stimulated symmetry breaking to induce ferroelectricity in a material and excite drug delivery or stimulate cell growth.

The concept is definitely interesting as the field of biocompatible ferroelectrics is rapidly growing. However, the coupling of optical and ferroelectric properties not so uncommon, and the authors themselves acknowledge that there have been similar phenomena observed in similar compounds with statements such as, "Schiff bases are well-known photochromic/thermochromic materials for constructing optically controlled ferroelectrics", which seems to dilute the claim of novelty for the work.

The methods used and the analysis seems mostly reasonable however some lacking of experimental method description and analysis leaves some parts of the work unclear. For example: With regards to figure 4 – concerning PFM. The results clearly show ferroelectric switching with electric field before and after UV light illumination (in enol and keto forms). This means the symmetry breaking is also induced by electric fields, but this phenomena is not discussed. The question is then how stable is this the ferroelectric state after switching

with either stimulus? The authors try to address this with Figure 5, but its not clear how this experiment was conducted. The authors describe that 5a and 5e show “Initial domain pattern in the dark”. What is the history of this sample? If it has not seen an electric field or light before why is there any domain pattern at all? It should be centrosymmetric. This experiment raises additional questions, for instance why does the domain pattern return after visible light irradiation with the same pattern as the initial state? If the symmetry is removed and then broken again why would the domain pattern be the same?

The manuscript also includes some rather confusing statements. One example is this “It should be highlighted that the enantiomeric organosilicon Schiff base compounds have photo-mediated bistability in multiple channels such as dielectric, ferroelectric polarization, and SHG properties in addition to ferroelectric bistability.” What is meant by “multiple channels”? It is well known that symmetry breaking changes the SHG signal, induces a spontaneous ferroelectric polarization and will change the dielectric response. These are not “in addition” to ferroelectric bistability but simply a result of it. So it seems to me the authors are claiming special properties that are in addition to the symmetry breaking when in fact they are necessary biproducts of it.

Finally, the manuscript is full of “over writing” which distracts the reader from the core topic and confuses the scientific message. Examples such as “molecular orbital breaking, which is a pearl drop in the ocean,” and “Symmetry breaking is widely found in nature from the birth of life to the mystery of the universe. In physics, it is the key to ushering in a new era...” this kind of thing just distracts from the message and value of the manuscript.

In summary, while I find this work fascinating and see that experiments have been performed with sufficient rigor, it is not clear in the present manuscript that the manuscript represents a high enough degree of novel information for Nature Comm, as the scientific observations, specifically related to the ferroelectric behavior, are not clearly enough explained.

Additional comments

1. Figure 2 has no e), but this is written I the caption.
2. Insufficient detail for how the samples were prepared for dielectric measurements – what electrodes were used?

Reviewer #3 (Remarks to the Author):

The authors reported a pair of organosilicon Schiff base ferroelectric crystals R- and S-SEA-Si with multiple photo-mediated properties including SHG, dielectricity, and ferroelectricity. They carefully investigated the mechanism of the photo-induced phase transition through crystal structure, molecular orbital, and other experiments. Since the light source is safe, clean, simple to operate, and easy to remotely control, R- and S-SEA-Si have broad application prospects in the biomedical field. They also verified the biocompatibility of the as-synthesized crystals indicating its potential as an implanted electronic device. When I read this manuscript, I felt very excited because it opened a new idea for investigation on the phase transition mechanism of ferroelectrics. One of the common characteristics of phase transition is the change in symmetry. Landau introduced symmetry breaking into phase transition theory and ferroelectric phase transition is caused by the breaking of spatial symmetry. According to Landau's phase transition theory and Curie's principle, there exists a parent-child relationship between the point groups of crystals before and after ferroelectric phase transition. These theories support the study of ferroelectric phase transition mechanisms. However, as more and more types of ferroelectric materials are being developed, these traditional theories find it difficult to explain all phenomena. For example, emerging photochromic ferroelectrics undergo ferroelectric-ferroelectric phase transitions under light, but these phase transitions are difficult to understand by symmetry breaking. The concept of molecular orbital breaking has been mentioned in some excellent works but they have not pointed out the specific application scenarios for these materials (Adv. Mater. 2023, 35, 2305471; Angew. Chem. Int. Ed. 2023, 62, e202315189). This article provides a detailed study of molecular orbital breaking and presents a paradigm for the phase transition mechanism of photoferroelectricity. The article also proposes a relatively complete system and tries to call on other scholars to pay attention to theoretical research of ferroelectrics at the electronic level. The proposal of new phase transition theories has extraordinary significance and will attract broad interests and wide readerships in the fields of material science, chemistry, biology, medicine, physics, and electronics. Therefore, I recommend the publication of this article in Nature Communications after minor revisions.

1. For the PFM measurement, how about the thickness of the crystalline thin film of R-SEA-Si.

2.The conditions of TGA measurement could be provided.

3.The authors employed mouse calvarial pre-osteoblast cell line MC3T3-E1 cells for in vitro investigations. While these cells are indeed valuable for osteogenic-related studies, it might be advantageous to consider stem cells, such as mesenchymal stem cells.

4.The authors are suggested to double-check the word spelling mistakes, such as 'trans-keno form' in Figure 3d. Applied light wavelengths should be consistent, with 448 nm shown in Scheme 1 but 488 nm denoted in other images.

Reviewer(s)' Comments to Author:

Reviewer: 1

Comments:

In this study, the authors report the synthesis of a pair of organosilicon Schiff base ferroelectric crystals, (*R*)- and (*S*)-*N*-(3,5-di-*tert*-butylbenzylidene)-1-((triphenylsilyl)oxy)ethanamine, which exhibit optically controlled phase transitions involving molecular orbital breaking. The cleavage of molecular orbitals indicates the cleavage and reconfiguration of covalent bonds (C=N and C=O conversion in the enol form and C=N and C=O conversion in the keto form) during the phase transition. In particular, the authors show that these organosilicon Schiff base compounds exhibit excellent biocompatibility and promote cell proliferation of MC3T3-E1 cells, as confirmed by cell viability and adhesion studies. I found this process reported by the authors interesting because it could lead to light-mediated bistability with multiple physical channels, including dielectricity, second harmonic generation, and ferroelectric polarization. On the other hand, I would like to request a clearer description of the following points.

Response: We sincerely thank the reviewer for spending his/her valuable time on carefully reviewing our manuscript, the in-depth comments, as well as his/her compliments on the impact and significance of our work. It is a great opportunity for us to communicate with excellent reviewers and learn from them a lot.

1. I thought the biocompatibility of organosilicon Schiff base compounds was an important result in this paper. The authors show their results in Figure 6, which I thought was an important result. On the other hand, biocompatibility testing is defined by international standards and is performed in accordance with ISO10993-5 and ISO10993-6. I would like to ask the authors to describe whether their biocompatibility tests are consistent with these results.

Response: We thank the reviewer for the valuable comments and suggestions. ISO 10993-5 furnishes comprehensive guidance on the assessment of cytotoxicity in medical samples, offering both qualitative and quantitative methodologies for discerning cell viability and detecting any potential adverse effects. In our investigation, we employed the CCK8 assay to assess cell viability. Furthermore, live/dead and phalloidin/DAPI staining were conducted to evaluate cell proliferation and cell adhesion of MC3T3-E1 cells, respectively, in order to ascertain any detrimental impacts on these cells. Thus, the biocompatibility assessments conducted in our study were aligned with the principles outlined in ISO 10993-5.

On the contrary, ISO 10993-6 delineates both qualitative and quantitative methodologies for assessing local effects subsequent to implantation, encompassing histological examination and biochemical analysis. To ascertain their biological safety, a 10-milligram dose of *R*-SEA-Si was subcutaneously implanted beneath the dorsal skins of mice for 3 days. After that, mouse serum was collected for biochemical analysis. Additionally, major organs (heart, liver, spleen, lung, and kidney) as well as the skin directly above the *R*-SEA-Si implant site were harvested for histological scrutiny. The analysis of serum parameters, including aspartate aminotransferase (AST), urea, and cholesterol (CHO), all fell within the normal range (Fig. 6g). Furthermore, neither major organs (Fig. 6h) nor the skin tissue directly overlying the *R*-SEA-Si implant site (Fig. 6i) displayed observable inflammatory cell infiltration or pathological characteristics.

Fig. 6 Biocompatibility assessment of *R*-SEA-Si. (g) Serum biochemical test results from control and *R*-SEA-Si administered mice at 3 days after subcutaneously implantation ($n \geq 5$ per group).

Fig. 6 Biocompatibility assessment of R-SEA-Si. (h) H&E staining images of mouse skin tissue directly above the implanted R-SEA-Si.

Fig. 6 Biocompatibility assessment of R-SEA-Si. (i) H&E staining images of major organs collected from control and R-SEA-Si administered mice at 3 days after subcutaneously implantation.

2. The authors' material is shown to have two states, with covalent bond cleavage and reconstitution during the phase transition process (Figure 2). On the other hand, the stability in each state has not been sufficiently demonstrated experimentally. I believe it is necessary to demonstrate stability in each state, such as stability over time, stability under ambient light in the atmosphere and other environments, and stability with respect to temperature.

Response: We thank the reviewer for instructive suggestions. Following your kind comment, we have investigated the stability of the *trans*-keto form of R-SEA-Si under dark and ambient light conditions by solid-state UV-vis spectra. As shown in Supplementary Fig. 6, the absorption intensity around 480 nm in *trans*-keto form

gradually weakened and nearly dropped back within 1 h under natural light, suggesting that the recovery from *trans*-keto form to enol form in *R*-SEA-Si. This process takes a longer time in the dark. According to the TGA results, the *trans*-keto form of *R*-SEA-Si possessed good thermal stability before 580 K, which is similar to that of the enol form. The corresponding figures and descriptions have been added to the revised manuscript.

The corresponding changes in the revised manuscript:

We have investigated the stability of the *trans*-keto form of *R*-SEA-Si through time-dependent solid-state UV-vis spectra and TGA measurements (Supplementary Figs. 6–7).

Supplementary Fig. 6 The time-dependent UV-vis spectra of *R*-SEA-Si irradiated with 365 nm for 1 hour: (a) in dark; (b) under natural light. The absorption intensity around 480 nm in *trans*-keto form gradually weakened and nearly dropped back within 1 h under natural light, suggesting that the recovery from *trans*-keto form to enol form in *R*-SEA-Si.

Supplementary Fig. 7 TGA curve of *R*-SEA-Si irradiated with 365 nm for 1 hour.

3. As a non-academic comment, it is unnatural for the Discussion chapter to begin with "In conclusion," which is not appropriate. In this paper, the Discussion is not sufficiently thorough. In particular, I think that the novelty should be clearly indicated again in terms of what points are novel compared to previous organosilicon Schiff base ferroelectric crystals. One perspective would be a novel application utilizing biocompatibility, but I do not believe that the biocompatibility tests have been shown from various experimental results. I think that standard experimental results to demonstrate biocompatibility should be presented, and I would like to request the presentation of experimental results that many researchers such as ISO10993, HE, Iba1, GFAP, etc. have shown as biocompatibility tests.

Response: We thank the reviewer for reminding us for that. We have deleted 'in conclusion' and improved the discussion according to the reviewer's suggestions. Many complementary experiments were performed on the basis of our initial study. To further investigate the potential biological toxicity of *R*-SEA-Si, human bone marrow-derived mesenchymal stem cells (hBMSCs) were utilized. Following a 48-hour exposure to the leach liquor of *R*-SEA-Si, hBMSCs exhibited robust cell viability (Supplementary Fig. 10), with no discernible effects on cell proliferation (Supplementary Fig. 11) or cell adhesion (Supplementary Fig. 12). In addition, the neurotoxin of *R*-SEA-Si was further investigated using rat neural stem cells (rNSCs) as models. Immunostaining images for

anti-beta III tubulin (Tuj-1) and anti-glia fibrillary acidic protein (GFAP) demonstrated that rNSCs treated with the leach liquor of *R*-SEA-Si could differentiate normally as in the control group (Supplementary Fig. 13). Collectively, these findings underscore the excellent biocompatibility of *R*-SEA-Si.

In addition, to ascertain their biological safety, a 10-milligram dose of *R*-SEA-Si was subcutaneously implanted beneath the dorsal skins of mice for 3 days. After that, mouse serum was collected for biochemical analysis. Additionally, major organs (heart, liver, spleen, lung, and kidney) as well as the skin directly above the *R*-SEA-Si implant site were harvested for histological scrutiny. The analysis of serum parameters, including aspartate aminotransferase (AST), urea, and cholesterol (CHO), all fell within the normal range (Fig. 6g). Furthermore, neither major organs Fig. 6h nor the skin tissue directly overlying the *R*-SEA-Si implant site (Fig. 6i) displayed observable inflammatory cell infiltration or pathological characteristics.

Supplementary Fig. 8 Cell viability of hBMSCs after incubation with the leach liquor of *R*-SEA-Si for 48 hours (n=6 per group).

Supplementary Fig. 9 Representative scan of live/dead staining of hBMSCs after treated with leach liquor of *R*-SEA-Si for 48 hours, green and red labeled cells represent living and dead cells, respectively.

Supplementary Fig. 12 Representative scan of phalloidine/DAPI staining of hBMSCs after treated with leach liquor of *R*-SEA-Si for 48 hours, red and blue labeled cells represent cytoskeleton and nucleus of cells, respectively.

Fig. 6 Biocompatibility assessment of R-SEA-Si. (g) Serum biochemical test results from control and R-SEA-Si administered mice at 3 days after subcutaneously implantation ($n \geq 5$ per group).

Fig. 6 Biocompatibility assessment of R-SEA-Si. (h) H&E staining images of mouse skin tissue directly above the implanted R-SEA-Si.

Fig. 6 Biocompatibility assessment of R-SEA-Si. (i) H&E staining images of major organs collected from control and R-SEA-Si administered mice at 3 days after subcutaneously implantation.

The corresponding changes in the revised version:

Discussion

Besides, these organosilicon compounds not only show good *in vitro* biocompatibility as well as a significant promotion effect on cell proliferation but also exhibit excellent *in vivo* biosafety when subcutaneously implanted into mice.

Biocompatibility of R-SEA-Si. To further investigate the potential biological toxicity of R-SEA-Si, human bone marrow-derived mesenchymal stem cells (hBMSCs) were utilized. Following a 48-hour exposure to the leach liquor of R-SEA-Si, hBMSCs exhibited robust cell viability (Supplementary Fig. 10), with no discernible effects on cell proliferation (Supplementary Fig. 11) or cell adhesion (Supplementary Fig. 12). In addition, the neurotoxin of R-SEA-Si towards rat neural stem cells (rNSCs) was further investigated. Immunostaining images for anti-beta III tubulin (Tuj-1) and anti-glia fibrillary acidic protein (GFAP) demonstrated that rNSCs treated with the leach liquor of R-SEA-Si could differentiate normally as in the control group (Supplementary Fig. 13). The *in vivo* biosafety of R-SEA-Si was further investigated by subcutaneously implantation of R-SEA-Si (10 mg) into ICR mice, the process of which was schematically shown in Fig. 6f. After 3 days of implantation, the analysis of serum parameters, including aspartate aminotransferase (AST), urea, and cholesterol (CHO), all fell within the normal range (Fig. 6g). Furthermore, neither the skin tissue directly overlying the R-SEA-Si implant site (Fig. 6h) nor the major organs (Fig. 6i) displayed observable inflammatory cell infiltration or pathological characteristics.

Fig. 6 Biocompatibility assessment of *R*-SEA-Si. (a) Schematic of *in vitro* biocompatibility assessment of *R*-SEA-Si towards MC3T3-E1 cells. (b) Cell viability of MC3T3-E1 cells after incubation with leach liquor of *R*-SEA-Si for 24 and 48 hours, respectively (n=4 per group). (c) Representative scan of live/dead staining of MC3T3-E1 cells after treated with leach liquor of *R*-SEA-Si for 24 and 48 hours, green and red labeled cells represent living and dead cells, respectively. (d) Quantification of live/dead staining in Fig. 6c using ImageJ. (e) Representative scan of phalloidine/DAPI staining of MC3T3-E1 cells after treated with leach liquor of *R*-SEA-Si for 24 and 48 hours, green and blue labeled cells represent cytoskeleton and nucleus of cells, respectively. (f) Schematic of *in vivo* biosafety assessment of *R*-SEA-Si towards ICR mice. (g) Serum biochemical test results from control and *R*-SEA-Si administered mice. (h) Histological images of skin from control and *R*-SEA-Si administered mice. (i) Histological images of major organs (Heart, Liver, Spleen, Lung, Kidney) from control and *R*-SEA-Si administered mice.

at 3 days after subcutaneously implantation ($n \geq 5$ per group). (h) H&E staining images of mouse skin tissue directly above the implanted *R*-SEA-Si. (i) H&E staining images of major organs collected from control and *R*-SEA-Si administered mice at 3 days after subcutaneously implantation.

***In vitro* biocompatibility assessment.** Mouse calvarial pre-osteoblast cell line MC3T3-E1 cells and human bone marrow-derived mesenchymal stem cells (hBMSCs) were obtained from the Institute of Life Science Cell Culture Center (Shanghai, China). Rat neural stem cells were isolated according to our previous study.

***In vivo* biosafety assessment.** 24 male ICR mice were obtained from the Laboratory Animal Center of Nanjing Drum Tower Hospital, Affiliated Hospital of Medical School, Nanjing University (China). All the experimental protocols in this study were approved by the Committee of Nanjing Drum Tower Hospital, Affiliated Hospital of Medical School, Nanjing University. A 10-milligram dose of *R*-SEA-Si was subcutaneously implanted into the beneath of the mouse dorsal skin. After 3 days of implantation, mouse serum was collected for serum biochemical analysis. Additionally, major organs (heart, liver, spleen, lung, and kidney) as well as the skin directly above the *R*-SEA-Si implant site were harvested for histological examination.

Reviewer: 2

Comments:

The article describes the investigation of molecular orbital breaking in organosilicon Schiff based molecular crystals as a novel foundation for ferroelectricity which can be induced by light stimulation. The orbital breaking mechanism involves the shift of a double bond in the compound from the C-N to the C-O bond. The primary interest seems to be the materials biocompatibility and thus the use of this light stimulated symmetry breaking to induce ferroelectricity in a material and excite drug delivery or stimulate cell growth.

Response: We would like to thank the reviewer for the valuable comments and suggestions. We also would like to thank him/her for the seriousness and carefulness in the reviewing process. According to the reviewers' comments and suggestions, the manuscript is revised in a very serious and deliberate way. With our tremendous efforts, we hope this revision will change reviewer 2's opinion.

The concept is definitely interesting as the field of biocompatible ferroelectrics is rapidly growing. However, the coupling of optical and ferroelectric properties not so uncommon, and the authors themselves acknowledge that there have been similar phenomena observed in similar compounds with statements such as, "Schiff bases are well-known photochromic/thermochromic materials for constructing optically controlled ferroelectrics", which seems to dilute the claim of novelty for the work.

Response: We do agree with the reviewer that the coupling of optical and ferroelectric properties is not so uncommon. We believe that optically controlled ferroelectrics are an important breakthrough in next-generation ferroelectric materials. Based on ferroelectrochemistry, many optically controlled ferroelectrics have rapidly sprung up (*J. Am. Chem. Soc.* 2021, 143, 13816–13823; *J. Am. Chem. Soc.* 2021, 143, 21685–21693; *J. Am. Chem. Soc.* 2022, 144, 19, 8633–8640; *Nat. Commun.* 2022, 13, 2379; *JACS Au* 2023, 3, 5, 1464–1471; *Angew. Chem. Int. Ed.* 2023, 62, e202315189;

Phys. Rev. Lett. 2023, 130, 176802; *J. Am. Chem. Soc.* 2021, 143, 21685–21693). Nevertheless, to date, the discovery of optically controlled ferroelectrics is still limited.

The novelty of our work can be concluded as follows: 1) This is the first discovery of a pair of organosilicon Schiff base ferroelectric crystals. Organosilicon ferroelectric combines the electromechanical conversion performance of ferroelectrics with the advantage of organosilicon materials. They have the advantages of flexibility, strong film-forming ability, nontoxicity, good biocompatibility, and acoustic impedance matching the human body. These unique features make organosilicon ferroelectrics more suitable for applications in personal health management, sports detection, and electronic skin than other conventional ferroelectrics (*Nat. Mater.* 2008, 7, 357–366). However, because of the lack of rational and efficient design strategies, the reported organosilicon ferroelectric compounds are rare to date; most importantly, all of them only exhibit ferroelectricity instead of other intriguing physical properties. In our work, we reported the first case of organosilicon Schiff base ferroelectric crystals, and they have photo-mediated bistability with multiple physical channels such as dielectric, second-harmonic generation, and ferroelectric polarization. 2) We used the molecular orbital breaking to interpret the phase transition mechanism from the perspective of electronics in the Schiff base ferroelectric system. Molecular orbital breaking can result in switchable structural and physical bistability in ferroelectric materials as traditional spatial symmetry breaking does. Differently, molecular orbital breaking interprets the phase transition mechanism from the perspective of electronics and sheds new light on manipulating the physical properties of ferroelectrics. In this work, the bistable switching is realized through reversible enol-keto photoisomerization. The mechanism of this process and the accompanied molecular orbital breaking are thoroughly studied. The variation of the orbits of N and O atoms and the breaking and recombination of covalent bonds support the photo-mediated transition. 3) Photo-mediated organosilicon Schiff base ferroelectrics with good biocompatibility are essential for bioelectronics because they can realize the modulation of physical properties in a moderate environment compared to conventional thermodynamics ferroelectrics. Contrary to conventional ferroelectrics whose physical properties are mediated by thermodynamics,

photo-mediated ferroelectric materials can realize the modulation in a moderate environment. The temperature-controlled ferroelectrics usually have extremely high phase transition temperature that exceeds the tolerance of human beings. The high temperature makes the modulation of physical properties a threat to human safety. The modulation of photo-mediated ferroelectrics can be realized by irradiation of specific light in the range of temperatures that people can adapt to, making them promising candidates for applications in personal health management, sports detection, electronic skin, and other fields (*Coord. Chem. Rev.* 2021, **447**, 214166).

For example, emerging photochromic ferroelectrics undergo ferroelectric-ferroelectric phase transitions under light, but these phase transitions are difficult to understand by symmetry breaking. The concept of molecular orbital breaking has been mentioned in some excellent works but they have not pointed out the specific application scenarios for these materials (*Adv. Mater.* 2023, 35, 2305471; *Angew. Chem. Int. Ed.* 2023, 62, e202315189).

Response: With the increasing emergence of photo-mediated ferroelectrics, a new mechanism of ferroelectrics–molecular orbit breaking has been proposed because of the lack of efficient theory to describe them (*Adv. Mater.* 2023, 35, 2305471). In 2023, this mechanism was first proposed in a photochromic organic diarylethene crystal DAE-1 (*Adv. Mater.* 2023, 35, 2305471). By alternating ultraviolet/visible light irradiation, DAE-1 experiences a reversible photoisomerization between two structural bistable states of closed-ring and open-ring forms, during which the molecular orbitals change with conversion between π and σ bonds, that is $3\pi \leftrightarrow 2\pi + \sigma$ in the molecule (Scheme 1). Light-induced molecular orbital breaking also causes the spatial symmetry breaking in lattice from the space group $Pca2_1$ to Pn . The coupling of electron spin, molecular orbital breaking, and spatial symmetry breaking in an asymmetric environment would offer more functional possibilities for data encryption and anti-counterfeiting.

[figure redacted]

Scheme 1. in the paper (Adv. Mater. 2023, 35, 2305471). The diagram of the molecular orbital conversion between photoisomers of DAE-1.

Later, a homochiral fulgide organic ferroelectric crystal (*E*)-(*R*)-3-methyl-3-cyclohexylidene-4-(diphenylmethylene)dihydro-2,5-furandione (**1**) was synthesized, which exhibits both ferroelectricity and photoisomerization. Importantly, the delocalized π -electronics portion in **1** shows the photoinduced reversible change of molecular orbitals from the 3 π molecular orbitals in open-ring isomer to 2 π and 1 σ molecular orbitals in closed-ring isomer, which enables the optical control of ferroelectric polarization. As far as we know, this is the first report revealing the manipulation of ferroelectric polarization in homochiral ferroelectric crystal by the photoinduced breaking of molecular orbitals.

[figure redacted]

Scheme 1. in the paper (Angew. Chem. Int. Ed. 2023, 62, e202315189). The molecular orbital breaking in optically controlled ferroelectric **1**.

As mentioned by the reviewer “the concept of molecular orbital breaking has been mentioned in some excellent works but they have not pointed out the specific application scenarios for these materials”, **the above-mentioned two cases of molecular orbit breaking do not systematically summarize the possible categories of molecular orbit breaking or point out its applicable situation. In our work, we first mentioned that molecular orbit breaking is as important as spatial symmetry breaking.** Molecular orbital breaking can result in switchable structural and physical bistability in ferroelectric materials as traditional spatial symmetry breaking does. Differently, molecular orbital breaking interprets the phase transition mechanism from the perspective of electronics and sheds new light on manipulating the physical properties of ferroelectrics. Taking *R*- and *S*-SEA-Si as an example, the molecular orbital breaking is manifested as the breaking and reformation of covalent bonds during

the phase transition process, that is, the conversion between C=N and C–O in the enol form and C–N and C=O in the keto form. This process brings about photo-mediated bistability with multiple physical channels such as dielectric, second-harmonic generation, and ferroelectric polarization. Thus, we think the discussion about molecular orbital breaking in our work completes its theory and extends its application to the Schiff base ferroelectric system, and these have not been mentioned in the previous work. Most importantly, we believe this type of photo-mediated ferroelectrics shows great application potential in bioelectronics. Contrary to conventional ferroelectrics whose physical properties are mediated by thermodynamics, photo-mediated ferroelectric materials can realize the modulation in a moderate environment. The modulation of photo-mediated ferroelectrics can be realized by irradiation of specific light in the range of temperatures that people can adapt to, making them promising candidates for applications in personal health management, sports detection, electronic skin, and other fields (*Coord. Chem. Rev.* 2021, **447**, 214166).

The methods used and the analysis seems mostly reasonable however some lacking of experimental method description and analysis leaves some parts of the work unclear. For example: With regards to figure 4 – concerning PFM. The results clearly show ferroelectric switching with electric field before and after UV light illumination (in enol and keto forms). This means the symmetry breaking is also induced by electric fields, but this phenomena is not discussed.

Response: We thank the reviewer for reminding us for that. Detailed information on the experimental method description and analysis has been added in the revised manuscript. The reviewer seems to have some misunderstanding of Figure 4. Figures 4a–c are the ferroelectric domain of the *R*-SEA-Si. Figures 4d–g are the box-in-box switched domain patterns, which indicate the ferroelectricity of *R*-SEA-Si. Figure 4h shows the PFM amplitude and phase loops by PFM switching spectroscopy, confirming the electrically controlled ferroelectric switching for *R*-SEA-Si. These PFM measurement results provide solid evidence for the ferroelectricity of *R*-SEA-Si. All of them were captured or measured on the crystalline thin film of *R*-SEA-Si in the enol

form, and this state is more stable in the ambient condition. After the light irradiation of 365 nm on the crystalline thin film of *R*-SEA-Si, the PFM amplitude and phase loops also present the typical butterfly and rectangle shapes, as shown in Figure 4i. This indicates that the *R*-SEA-Si in the keto form also shows ferroelectricity. It can be found that the two states (mediated by the light irradiation) have different ferroelectric switching features such as the coercive field. However, the electric field would not cause symmetry breaking. The electric field is one way to achieve ferroelectric polarization switching, but it would not result in the transition of *R*-SEA-Si between the enol and keto forms to our knowledge.

The question is then how stable is this the ferroelectric state after switching with either stimulus? The authors try to address this with Figure 5, but its not clear how this experiment was conducted.

Response: Figure 5 is provided to demonstrate the photo-mediated ferroelectricity. To investigate the optical control of the domains in *R*-SEA-Si, the experiment was conducted as follows. Firstly, we imaged the out-of-plane PFM phase and amplitude image of the initial domain pattern in the dark, corresponding to the enol state (Fig. 5a and 5d). Then, the UV light irradiation of 365 nm was exposed to the surface of the *R*-SEA-Si crystalline thin film, and we in situ recorded the PFM images. As shown in Fig. 5b and 5e, the optical poling domain structure of *trans*-keto state was obtained upon UV light irradiation of 365 nm, resulting in a uniform brown single-domain state. After that, the visible light of 488 nm was exposed to the sample surface (Fig. 5c and 5f). The brown domain can be recovered to light green domains after laser exposure of 488 nm, corresponding to the enol state. The light sources utilized here are LED lamps with a rated power of 3000 mW and 60 mW for wavelengths of 365 nm and 488 nm, respectively. We have to admit that the ferroelectric state after optical poling is not so stable because the enol state of this Schiff base compound is much more stable than that of the keto state. To visualize the ferroelectric domain in the keto state, we should keep the irradiation of UV light toward the sample surface. When we remove the UV light, the domain would change into the initial state corresponding to the enol state.

The corresponding changes in the revised version:

To investigate the optical control of the domains in *R*-SEA-Si, the experiment was conducted as follows. Firstly, we imaged the out-of-plane PFM phase and amplitude image of the initial domain pattern in the dark, corresponding to the enol state (Fig. 5a and 5d). Then, the UV light irradiation of 365 nm was exposed to the surface of the *R*-SEA-Si crystalline thin film, and we in situ recorded the PFM images. As shown in Fig. 5b and 5e, the optical poling domain structure of *trans*-keto state was obtained upon UV light irradiation of 365 nm, resulting in a uniform brown single-domain state. After that, the visible light of 488 nm was exposed to the sample surface (Fig. 5c and 5f). The brown domain can be recovered to light green domains after laser exposure of 488 nm, corresponding to the enol state. The light sources utilized here are LED lamps with a rated power of 3000 mW and 60 mW for wavelengths of 365 nm and 488 nm, respectively.

The authors describe that 5a and 5e show “Initial domain pattern in the dark”. What is the history of this sample? If it has not seen an electric field or light before why is there any domain pattern at all? It should be centrosymmetric.

Response: The polycrystalline thin film of *R*-SEA-Si used for the PFM measurement is fabricated by the drop-casting method. Thus, the initial domain pattern in the dark was recorded on the crystalline thin film of *R*-SEA-Si in the enol form. This domain region has not been irradiated by the UV light.

Because of the spontaneous polarization of ferroelectric crystal, positive and negative bound charges exist at either end, respectively. The electric field generated by bound charges is opposite to polarization inside the crystal (known as depolarization field), causing an increase in electrostatic energy. When subjected to mechanical constraints, the strain accompanied by spontaneous polarization will also increase the strain energy. Thus, the state of uniform polarization is unstable, and the crystal will be divided into several small regions. The electric dipoles inside each small region toward the same direction, but the direction of the electric dipoles in each small region is different. These small areas are called electric domains. Therefore, the ferroelectric

domain forms in order to decrease the electrostatic energy and strain energy. Generally speaking, the ferroelectric domain structure depends strongly on the symmetry of the paraelectric phase and the orientation of spontaneous polarization (*Cryst. Res. Technol.* 2006, 41, 1045–1048). The symmetry of the single crystal is equal to that of the ferroelectric phase. In their paraelectric phase, there exist several directions corresponding to the direction of spontaneous polarization in the ferroelectric phase, and the probability of spontaneous polarization emerging along these directions is equal (*Chem. Soc. Rev.* 2021, 50, 8248–8278). This results in the formation of a ferroelectric multi-domain structure on the crystal. That is why the domains exist in their virgin state.

The statement of the reviewer “it should be centrosymmetric” is wrong. From the perspective of crystallographic structure, we have used single-crystal X-ray diffraction measurement to identify its polar (let alone non-centrosymmetric) structure. The relationships between macroscopic symmetries and macroscopic physical properties of crystals are governed by Neumann’s principle, which states that the symmetry of any physical property of a crystal must include all the symmetry elements of the point group of the crystal (*Chem. Soc. Rev.* 2016, 45, 3811–3827). Inspired by Neumann’s principle, a second harmonic generation (SHG) technique has been developed as an indispensable supplementary approach for the confirmation of polar structure. *R*-SEA-Si is a SHG-active material, verifying its non-centrosymmetric structure. Further, PFM measurements also provide solid evidence for the polar structure of *R*-SEA-Si. Thus, *R*-SEA-Si should be a ferroelectric compound with a polar structure instead of a centrosymmetric one.

This experiment raises additional questions, for instance why does the domain pattern return after visible light irradiation with the same pattern as the initial state? If the symmetry is removed and then broken again why would the domain pattern be the same?

Response: We thank the reviewer for the good comment. It is known that the domain structure is closely related to the crystal symmetry. Specifically, the ferroelectric domain structure depends strongly on the symmetry of the paraelectric

phase and the orientation of spontaneous polarization (*Chem. Soc. Rev.* 2021, 50, 8248–8278). For example, Im-ClO₄ with the $\bar{3}mF3m$ phase transition is a typical uniaxial ferroelectric, which reveals the virgin hexagonal-type domains in both the single crystal and the thin film, similar to those observed in LiNbO₃ (Fig. 10d).

[figure redacted]

Figure 10d. in the paper (*Chem. Soc. Rev.*, 2021, 50, 8248–8278). Vertical PFM phase image for Im-ClO₄.

Photo-mediated ferroelectrics generally undergo photoinduced geometrical isomerization such as *trans-cis* or enol-keto isomerization, which can trigger a type of photoinduced structural phase transition different from a thermodynamic transition (*J. Am. Chem. Soc.* 2021, 143, 34, 13816–13823). These two photo-triggered ferroelectric states have their corresponding domain pattern. Theoretically speaking, if this crystal does not show any damage, the domain structures of these two states should be the same. That is why the domain pattern after the irradiation of 488 nm is the same as that of the initial state. Similar phenomena have been found in other photo-mediated ferroelectrics. Taking SA-NPh-(*R*) and SANPh-(*S*) as an example, the native ferroelectric/ferroelastic domain patterns (State A) can completely disappear (State B) by 365 nm UV irradiation and reversibly reemerge (State A') upon 488 nm visible light irradiation (Fig. 4) (*Nat. Commun.* 2022, 13, 2379). Most importantly, the domain patterns in the final state (State A') are the same as in the original state (State A), demonstrating the presence of memory effect for the ferroic domains in these compounds.

[Editorial note: Citation for the figure below.

Wang, ZX., Chen, XG., Song, XJ. *et al.* Domain memory effect in the organic ferroics. *Nat Commun* **13**, 2379 (2022). <https://doi.org/10.1038/s41467-022-30085-1>]

Fig. 4. in the paper (*Nat. Commun.*, 2022, 13, 2379). Ferroelectric domains change by heat and light. Consecutive images showing erasure and recurrence of stripe domains in SA-NPh-(S) thin-film via both thermal treatment and light irradiation. The red and blue regions indicate the two different polarization-oriented states of ferroelectric domains. Domain patterns for SA-NPh-(S) thin-film (a) at the initial state, (b) after heating to Phase II and (c) cooling down to room temperature, subsequently (d) after 365 nm UV light irradiation, and (e) after 488 visible light irradiation.

The manuscript also includes some rather confusing statements. One example is this “It should be highlighted that the enantiomeric organosilicon Schiff base compounds have photo-mediated bistability in multiple channels such as dielectric, ferroelectric polarization, and SHG properties in addition to ferroelectric bistability.” What is meant by “multiple channels”?

Response: The multiple channels here indicate the bistability which can be realized in multiple physical properties. This description has been used in the research article

published in *Advanced Materials* (*Adv. Mater.* 2014, 26, 4515–4520). In that article, the authors mentioned that “studies on bistable materials have been recently extended to systems of molecular compounds which exhibit bistability simultaneously in multiple physical channels, such as optical, electrical, and magnetic, which would give extra freedom in device design” and “it exhibits simultaneously switchable bistability in three channels of dielectric, piezoelectric, and SHG properties in addition to ferroelectric bistability”. Similarly, we used photo-mediated bistability in multiple channels to describe these bistable properties.

It is well known that symmetry breaking changes the SHG signal, induces a spontaneous ferroelectric polarization and will change the dielectric response. These are not “in addition” to ferroelectric bistability but simply a result of it. So it seems to me the authors are claiming special properties that are in addition to the symmetry breaking when in fact they are necessary biproducts of it.

Response: We do agree with the reviewer that SHG intensity, spontaneous polarization, and dielectric constant are closely related to the ferroelectric state. Thus, light irradiation can realize the phase transition between two ferroelectric states, which corresponds to different physical properties such as SHG intensity, spontaneous polarization, and dielectric constant. That is how this photo-mediated bistability in multiple channels works. We also agree with the reviewer that these bistabilities are accompanied by ferroelectric bistability instead of the newly produced ones. However, we do realize the precise photo-modulation of multiple physical properties in this organosilicon Schiff base ferroelectric crystal, and this room-temperature mediation can support the application in bioelectronics. We thus believe this photo-mediated bistability with multiple physical channels such as dielectric, second-harmonic generation, and ferroelectric polarization is of great significance for application.

Finally, the manuscript is full of “over writing” which distracts the reader from the core topic and confuses the scientific message. Examples such as “molecular orbital breaking, which is a pearl drop in the ocean,” and “Symmetry breaking is widely found in nature from the birth of life to the mystery of the universe. In physics, it is the key to

ushering in a new era...” this kind of thing just distracts from the message and value of the manuscript.

Response: According to the reviewers’ comments and suggestions, the manuscript is revised in a very serious and deliberate way to avoid “overwriting”.

The corresponding changes in the revised version:

Molecular orbital breaking can result in switchable structural and physical bistability in ferroelectric materials as traditional spatial symmetry breaking does.

Symmetry breaking, which is widely found in nature, is the key to understanding many interesting physical phenomena^{1, 2, 3, 4}.

In summary, while I find this work fascinating and see that experiments have been performed with sufficient rigor, it is not clear in the present manuscript that the manuscript represents a high enough degree of novel information for Nature Comm, as the scientific observations, specifically related to the ferroelectric behavior, are not clearly enough explained.

Response: We have carried out new experiments, made substantial revisions to the manuscript and adequately addressed all the concerns proposed by the reviewers. With our tremendous efforts, we hope this revision would change reviewer 2’s opinion.

Significance:

Ferroelectric materials are widely used in various applications such as data storage, sensors, and transducers because of their robust spontaneous electrical polarization¹. They generally undergo phase transitions which can be described as spatial symmetry breaking by Landau’s phenomenological theory^{2, 3}. Phase transition is essential for ferroelectrics since it can bring about tunable physical properties⁴. The phase transition mechanism of ferroelectrics mainly includes two types: order-disorder and displacive type. The former is usually found in some molecular ferroelectrics, while the latter is common in inorganic ceramics such as BaTiO₃ and Pb(Ti, Zr)O₃⁵. These processes are accompanied by changes in symmetric elements, which is well-known as symmetry breaking. Symmetry breaking will contribute to the rearrangement of ferroic orders and

changes in spontaneous polarization, thereby achieving regulation of ferroelectric properties⁶.

Recently, a new type of ferroelectric material, photo-mediated ferroelectric whose spontaneous polarization can be switched reversibly with a photoinduced phase transition triggered by structural photoisomerization has been proposed. This is a breakthrough in next-generation ferroelectric materials because photoirradiation stands out as a nondestructive, noncontact, and remote-control mean beyond an electric or strain field⁷. Among them, the typical photochromic compounds, such as Schiff base, azobenzene, diarylethene derivative, and spiropyran, have been chosen for light-driven ferroelectric materials, whose structural change is generally caused by photoisomerization instead of the traditional thermodynamic structural phase transition^{8,9}. However, the conventional phase transition mechanism cannot be used to describe them. Thus, a new ferroelectric phase transition driven by switchable covalent bonds is proposed. Molecular orbital breaking can help us to understand this unconventional ferroelectric mechanism from the perspective of electronics, and further provides a new approach to modulating ferroelectric properties at the electronic level especially the polarization, which is a beneficial supplement to symmetry breaking. The core content of molecular orbital breaking is the valence bond recombination, for which the phase transition process involves the breaking and reformation of covalent bonds and thus leads to polarization changes. For example, Liao *et al.* disclosed the dual breaking of molecular orbitals and spatial symmetry in a photochromic diarylethene ferroelectric material¹⁰. This is the first time that this new ferroelectric mechanism—molecular orbital breaking is proposed. Meanwhile, such a photoinduced phase transition is entirely driven by switchable covalent bonds with breaking and reformation, enabling the reversible light-controllable ferroelectric polarization switching, dielectric, and nonlinear optical bistability. It should be highlighted that in comparison with traditional ones driven by thermal stimuli, photo-mediated ferroelectric materials will have promising application prospects in biomedicine since their physical properties can be mediated by light at a temperature matching the human body, making them promising candidates for applications in personal health

management, sports detection, electronic skin, and other fields¹¹.

Here, we synthesized a pair of organosilicon Schiff base ferroelectric crystals, (*R*)- and (*S*)-*N*-(3,5-di-*tert*-butylbenzylidene)-1-((triphenylsilyl)oxy)ethanamine. They show optically controlled phase transition accompanying the molecular orbital breaking. The molecular orbital breaking is manifested as the breaking and reformation of covalent bonds during the phase transition process, that is, the conversion between C=N and C–O in the enol form and C–N and C=O in the keto form. As expected, these organosilicon Schiff base compounds show good biocompatibility and promotion of cell proliferation in biological cells confirmed by the tests of cell viability and adhesion. These intriguing features enable the multi-channel modulation in physical properties without the changing of temperature, thus making *R*- and *S*-SEA-Si a beneficial supplement to traditional ferroelectrics whose switchable properties are generally realized by thermodynamics. Therefore, this finding extends the ferroelectric mechanism for the optically-controlled ferroelectrics and highlights the important and unique applications of this type of materials in biomedicine. This will attract the attention of scientists from multiple disciplines such as chemistry, physics, electronics, biomedicine, and materials science.

Highlights:

1. The discovery of a pair of organosilicon Schiff base ferroelectric crystals having photo-mediated bistability with multiple physical channels such as dielectric, second-harmonic generation, and ferroelectric polarization.
2. We first used molecular orbital breaking to interpret the phase transition mechanism from the perspective of electronics in the Schiff base ferroelectric system.
3. Photo-mediated organosilicon Schiff base ferroelectrics with good biocompatibility are essential for bioelectronics because they can realize the modulation of physical properties in a moderate environment compared to conventional thermodynamics ferroelectrics.

Additional comments

1. Figure 2 has no e), but this is written I the caption.

Response: We thank the reviewer for reminding us for that. We have deleted e) in the revised version.

The corresponding changes in the revised version:

Fig. 2 The spectra and orbital structures of *R*- and *S*-SEA-Si. Experimental UV-Vis (a) and IR absorption (b) spectra of *R*-SEA-Si under ambient conditions and after UV irradiation. HOMO (c) and LUMO (d) of the *R*-SEA-Si molecule.

2. Insufficient detail for how the samples were prepared for dielectric measurements – what electrodes were used?

Response: We thank the reviewer for reminding us for that. The dielectric measurements were carried out on a Tonghui TH2828A impedance analyzer. The capacitor for dielectric measurement was prepared by melting the sample between two ITO transparent glasses and being recrystallized at room temperature to form a sandwich-like structure. The area of electrodes and the thickness of the film are approximately 90 mm² and 30 μm, respectively. In the photo-triggered dielectric measurement, the capacitor was alternately illuminated by the 365 nm and 488 nm light from two sides in the dark.

The corresponding changes in the revised version:

Dielectric measurements. The dielectric measurements were carried out on a Tonghui TH2828A impedance analyzer. The capacitor for dielectric measurement was prepared by melting the sample between two ITO transparent glasses and being recrystallized at room temperature to form a sandwich-like structure. The area of electrodes and the thickness of the film are approximately 90 mm² and 30 μm, respectively. In the photo-triggered dielectric measurement, the capacitor was alternately illuminated by the 365 nm and 488 nm light from two sides in the dark.

Reviewer: 3

Comments:

The authors reported a pair of organosilicon Schiff base ferroelectric crystals R- and S-SEA-Si with multiple photo-mediated properties including SHG, dielectricity, and ferroelectricity. They carefully investigated the mechanism of the photo-induced phase transition through crystal structure, molecular orbital, and other experiments. Since the light source is safe, clean, simple to operate, and easy to remotely control, R- and S-SEA-Si have broad application prospects in the biomedical field. They also verified the biocompatibility of the as-synthesized crystals indicating its potential as an implanted electronic device. When I read this manuscript, I felt very excited because it opened a new idea for investigation on the phase transition mechanism of ferroelectrics. One of the common characteristics of phase transition is the change in symmetry. Landau introduced symmetry breaking into phase transition theory and ferroelectric phase transition is caused by the breaking of spatial symmetry. According to Landau's phase transition theory and Curie's principle, there exists a parent-child relationship between the point groups of crystals before and after ferroelectric phase transition. These theories support the study of ferroelectric phase transition mechanisms. However, as more and more types of ferroelectric materials are being developed, these traditional theories find it difficult to explain all phenomena. For example, emerging photochromic ferroelectrics undergo ferroelectric-ferroelectric phase transitions under light, but these phase transitions are difficult to understand by symmetry breaking. The concept of molecular orbital breaking has been mentioned in some excellent works but they have not pointed out the specific application scenarios for these materials (*Adv. Mater.* 2023, 35, 2305471; *Angew. Chem. Int. Ed.* 2023, 62, e202315189). This article provides a detailed study of molecular orbital breaking and presents a paradigm for the phase transition mechanism of photoferroelectricity. The article also proposes a relatively complete system and tries to call on other scholars to pay attention to theoretical research of ferroelectrics at the electronic level. The proposal of new phase transition

theories has extraordinary significance and will attract broad interests and wide readerships in the fields of material science, chemistry, biology, medicine, physics, and electronics. Therefore, I recommend the publication of this article in Nature Communications after minor revisions.

Response: We sincerely thank the reviewer for spending his/her valuable time on carefully reviewing our manuscript, the in-depth comments, as well as his/her compliments on the impact and significance of our work. We have addressed all problems and made substantial modifications to our manuscript.

1. For the PFM measurement, how about the thickness of the crystalline thin film of R-SEA-Si.

Response: We greatly appreciate the reviewer's comments. The thickness of the crystalline thin film of R-SEA-Si is about 1.53 μm . We have mentioned it in the revised version.

The corresponding changes in the revised version:

PFM characterization. The PFM measurement was carried out on a commercial piezoresponse force microscope (Oxford instrument, Cypher ES) with a high-voltage package, in-situ heating stage, and custom-designed light sources. Conductive Pt/Ir-coated silicon probes (EFM, Nanoworld) were used for domain imaging and polarization switching studies, with a nominal spring constant of ~ 2.8 nN/nm and a free-air resonance frequency of ~ 75 kHz. The thickness of the thin film of R-SEA-Si is about 1.53 μm .

2. The conditions of TGA measurement could be provided.

Response: We thank the reviewer for reminding us for that. We have mentioned it in the revised version.

The corresponding changes in the revised version:

The TGA measurement of S-SEA-Si, R-SEA-Si, and Rac-SEA-Si were performed on a PerkinElmer TGA 8000 at a heating rate of 30 K min^{-1} in the atmosphere.

3. The authors employed mouse calvarial pre-osteoblast cell line MC3T3-E1 cells for in vitro investigations. While these cells are indeed valuable for osteogenic-related studies, it might be advantageous to consider stem cells, such as mesenchymal stem cells.

Response: We thank the reviewer for the good comments. To further investigate the potential biological toxicity of *R*-SEA-Si, human bone marrow-derived mesenchymal stem cells (hBMSCs) were utilized. Following a 48-hour exposure to the leach liquor of *R*-SEA-Si, hBMSCs exhibited robust cell viability (see Supplementary Fig. 10), with no discernible effects on cell proliferation (see Supplementary Fig. 11) or cell adhesion (see Supplementary Fig. 12).

Supplementary Fig. 10 Cell viability of hBMSCs after incubation with the leach liquor of *R*-SEA-Si for 48 hours (n=6 per group).

Supplementary Fig. 11 Representative scan of live/dead staining of hBMSCs after treated with leach liquor of *R*-SEA-Si for 48 hours, green and red labeled cells represent living and dead cells, respectively.

Supplementary Fig. 12 Representative scan of phalloidine/DAPI staining of hBMSCs after treated with leach liquor of *R*-SEA-Si for 48 hours, red and blue labeled cells represent cytoskeleton and nucleus of cells, respectively.

4. The authors are suggested to double-check the word spelling mistakes, such as ‘trans-keno form’ in Figure 3d. Applied light wavelengths should be consistent, with 448 nm shown in Scheme 1 but 488 nm denoted in other images.

Response: We thank the reviewer for reminding us for that. According to the reviewer’s comments, we have carefully double-checked the word spelling mistakes and revised our manuscript in the revised version.

The corresponding changes in the revised version:

Fig. 3 Photo-mediated bistability of R-SEA-Si in three physical channels: dielectricity, SHG intensity, and ferroelectricity. (a) Comparison of the conventional temperature-controlled and photo-mediated bistability. (b) The real part ϵ' at room temperature between the irradiation of 365 and 488 nm. (c) SHG intensity between the irradiation of 365 and 488 nm at room temperature. (d) Schematic diagram of the molecular conformation transformation. The blue arrow indicates the direction of the molecular dipole.

Scheme 1. Schematic of three types of molecular orbital breaking.

REVIEWERS' COMMENTS

Reviewer #1 (Remarks to the Author):

The authors have responded courteously and appropriately to the reviewers' suggestions, and the paper has been extensively revised accordingly. The newly added experimental results, their description and discussion in the revised paper contain important content and make the paper better.

Based on overall considerations, I have determined that the revised paper is worthy of publication.

Reviewer #2 (Remarks to the Author):

I thank the authors for thoroughly considering the reviewers comments.

i think the authors have sufficiently addressed all comments and recommend publication of the manuscript.

Thank you.

Reviewer #3 (Remarks to the Author):

I have reached and carefully reviewed the point-to-point responses by the authors, and I believed that the current revision has been significantly improved. Before the final acceptance, a minor point in Scheme 1 that the authors may need to pay attention to. Since the photochromic molecular transformation from enol to trans-keto forms occurs without destroying the single-crystal form, it was assumed that the transformation occurs via pedal motion in a manner similar to the thermal motion of azobenzene crystals (Harada et al., *Acta Cryst.* 1997, B53, 662–672; Harada and Ogawa, *Chem. Soc. Rev.* 2009, 38, 2244–2252), which is known as a space-efficient motion in crystals. Therefore, the possibility of overall molecular rotation of the chiral SEA-Si group during the photoisomerization is very small. I think the authors can shortly address this minor issue and this paper could be accepted without further review with mine.

Reviewer #1 (Remarks to the Author):

The authors have responded courteously and appropriately to the reviewers' suggestions, and the paper has been extensively revised accordingly. The newly added experimental results, their description and discussion in the revised paper contain important content and make the paper better.

Based on overall considerations, I have determined that the revised paper is worthy of publication.

Response: We would like to express our gratitude to the reviewer for spending time and effort to help increase the impact of our work.

Reviewer #2 (Remarks to the Author):

I thank the authors for thoroughly considering the reviewers comments.

i think the authors have sufficiently addressed all comments and recommend publication of the manuscript.

Thank you.

Response: We thank the reviewer for his/her generally positive assessment of the work and the valuable suggestions.

Reviewer #3 (Remarks to the Author):

I have reached and carefully reviewed the point-to-point responses by the authors, and I believed that the current revision has been significantly improved. Before the final acceptance, a minor point in Scheme 1 that the authors may need to pay attention to. Since the photochromic molecular transformation from enol to trans-keto forms occurs without destroying the single-crystal form, it was assumed that the transformation occurs via pedal motion in a manner similar to the thermal motion of azobenzene crystals (Harada et al., Acta Cryst. 1997, B53, 662–672; Harada and Ogawa, Chem. Soc. Rev. 2009, 38, 2244–2252), which is known as a space-efficient motion in crystals. Therefore, the possibility of overall molecular rotation of the chiral SEA-Si group during the photoisomerization is very small. I think the authors can shortly address this minor issue and this paper could be accepted without further review with mine.

Response: We sincerely thank reviewer for spending his/her valuable time carefully reviewing our manuscript, the in-depth comments, as well as his/her compliments on the impact and significance of our work. We have revised the Scheme 1 as the reviewer suggested.